# Cell-scale biophysical determinants of cell competition in epithelia

**Daniel Gradeci[1,2†], Anna Bove[2,3†], Giulia Vallardi[4], Alan R Lowe[2,4,5]\*, Shiladitya Banerjee[1,5,6]\*, Guillaume Charras[2,3,5]\***

[1]Department of Physics and Astronomy, University College London, London, United Kingdom; [2]London Centre for Nanotechnology, University College London, London, United Kingdom; [3]Department of Cell and Developmental Biology, University College London, London, United Kingdom; [4]Institute for Structural and Molecular Biology, University College London, London, United Kingdom; [5]Institute for the Physics of Living Systems, University College London, London, United Kingdom; [6]Department of Physics, Carnegie Mellon University, Pittsburgh, United States

**Abstract** How cells with different genetic makeups compete in tissues is an outstanding question in developmental biology and cancer research. Studies in recent years have revealed that cell competition can either be driven by short-range biochemical signalling or by long-range mechanical stresses in the tissue. To date, cell competition has generally been characterised at the population scale, leaving the single-cell-level mechanisms of competition elusive. Here, we use high time-resolution experimental data to construct a multi-scale agent-based model for epithelial cell competition and use it to gain a conceptual understanding of the cellular factors that governs competition in cell populations within tissues. We find that a key determinant of mechanical competition is the difference in homeostatic density between winners and losers, while differences in growth rates and tissue organisation do not affect competition end result. In contrast, the outcome and kinetics of biochemical competition is strongly influenced by local tissue organisation. Indeed, when loser cells are homogenously mixed with winners at the onset of competition, they are eradicated; however, when they are spatially separated, winner and loser cells coexist for long times. These findings suggest distinct biophysical origins for mechanical and biochemical modes of cell competition.

**\*For correspondence:**
a.lowe@ucl.ac.uk (ARL);
shiladtb@andrew.cmu.edu (SB);
g.charras@ucl.ac.uk (GC)

[†]These authors contributed equally to this work

**Competing interests:** The authors declare that no competing interests exist.

## Introduction

Cell competition is a fitness control mechanism in which less fit cells (the losers) are eliminated from a tissue for optimal survival of the host (*Vincent et al., 2013*; *Levayer and Moreno, 2013*). First discovered in the *Drosophila* wing disc (*Morata and Ripoll, 1975*), cell competition has since been observed in many other physiological and pathophysiological contexts, especially in embryogenesis (*Amoyel and Bach, 2014*) and the development of tumours (*Chen et al., 2012*; *Madan et al., 2019*). While there have been extensive population-scale studies of competition (*Moreno et al., 2002*; *Wagstaff et al., 2016*), the competitive strategies and their underlying mechanisms at the level of single cells remain poorly understood.

Two broad conceptual classes of cell competition have been described. Mechanical competition arises because loser cells are more sensitive to crowding than winners (*Shraiman, 2005*). Losers are thought to die cell-autonomously because the overall cell density is increased by the growth of winners and, as a result, loser cells far from the interface with winners may die (*Wagstaff et al., 2016*; *Levayer et al., 2016*). By contrast, during biochemical competition, signalling occurs at the interface between cell types leading to apoptosis of loser cells only when in direct contact with winners (*Moreno et al., 2002*; *Yamamoto et al., 2017*). Here, the probability of elimination depends on the

extent of contact a loser cell has with the winners (*Levayer et al., 2015*; *Díaz-Díaz et al., 2017*). As a result, perturbations affecting the strength of intercellular adhesions strongly affect the outcome of competition, suggesting that cell mixing is an important factor in biochemical competition (*Levayer et al., 2015*).

One challenge in understanding cell competition from experimental data is that it takes place over several days, making the tracking of a cell's environment and its eventual fate challenging. The emergence of automated long-term microscopy and advanced image analysis for segmentation and cell state recognition enables hypotheses to be formulated regarding the mechanisms of cell elimination (*Gradeci et al., 2020*). For example, recent work has shown that loser cell death in an experimental model system for mechanical competition is strongly influenced by local cell density as expected, but that, in addition, division of winner cells appears favoured in neighbourhoods with many loser cells, something reminiscent of biochemical competition (*Bove et al., 2017*). Therefore, multiple modes of competition may be at play simultaneously and which of these determines the outcome remains unclear.

One way of gaining conceptual understanding into a complex multi-variate biophysical process is through computational or mathematical modelling. While population-scale models of competition based on ordinary or partial differential equations capture the overall behaviour of the tissue (*Shraiman, 2005*; *Bove et al., 2017*; *Nishikawa et al., 2016*), they do not provide insights into the influence of local tissue organisation, mechanics and cell-cell signalling on the outcome of competition. Cell-resolution computational models are well suited for describing how the behaviour of single cells and cell-cell interactions leads to population-scale dynamics (*Tsuboi et al., 2018*; *Lee and Morishita, 2017*). Although cell-scale models of cell competition have been developed (*Tsuboi et al., 2018*), they have not yet been used to test different competitive strategies or investigate the physical and topological parameters that are important in competition. This is partly due to the lack of high time-resolution experimental data to allow a robust comparison of models with experimental evidence as well as challenges in computationally implementing basic biological phenomena thought to be central to competition, such as the ability of epithelia to maintain a constant cell density (their homeostatic density) by attaining a balance of cell death and cell division.

Here, we develop a multi-scale agent-based computational model to gain conceptual understanding of the single-cell mechanisms that govern cell competition. Our modelling study is informed by our own experimental work in which we characterised single-cell mechanical competition using automatic annotation of movies lasting up to 4 days (*Bove et al., 2017*). Following analysis, these movies provide the fate and position of all cells over time, allowing for rigorous comparison of simulation to experiments. After calibrating the behaviour of winner and loser cells based on movies of pure cell populations, we show that we can replicate competition when the two distinct cell types are mixed and investigate the impact of each interaction and kinetic parameters on the outcome of mechanical cell competition. We then implement a model of biochemical competition based on contact-dependent death that can replicate all current experimental observations and uncover the key parameters influencing its outcome. We find that mechanical competition appears to be controlled by the difference in homeostatic density between cell types, whereas biochemical competition is governed by tissue organisation.

## Results

### Experimental pipeline

In our experiments, we examined competition between wild-type Madin–Darby Canine kidney epithelial cells (winners, MDCK$^{WT}$) and cells depleted for the polarity protein scribble (losers, MDCK$^{Scrib}$) (*Norman et al., 2012*). To allow for simple image analysis, each cell type stably expressed a histone marker fused to a different fluorophore (MDCK$^{WT}$:GFP and MDCK$^{Scrib}$:mRFP). Cells were seeded in various ratios of loser:winner cells (10:90, 50:50, 90:10) as well as colonies and imaged for up to 96 hr at 4 min intervals (*Figure 1A*). Cell segmentation and tracking allowed to determine population measurements (such as the evolution of cell count, the number of mitoses, and the number of apoptoses) as well as cellular-scale measurements (such as local cell density, number of neighbours, identity of neighbours, and cell state) for each cell type (*Bove et al., 2017*). These data provided the metrics to compare simulations to experiments (*Figure 1B, C*).

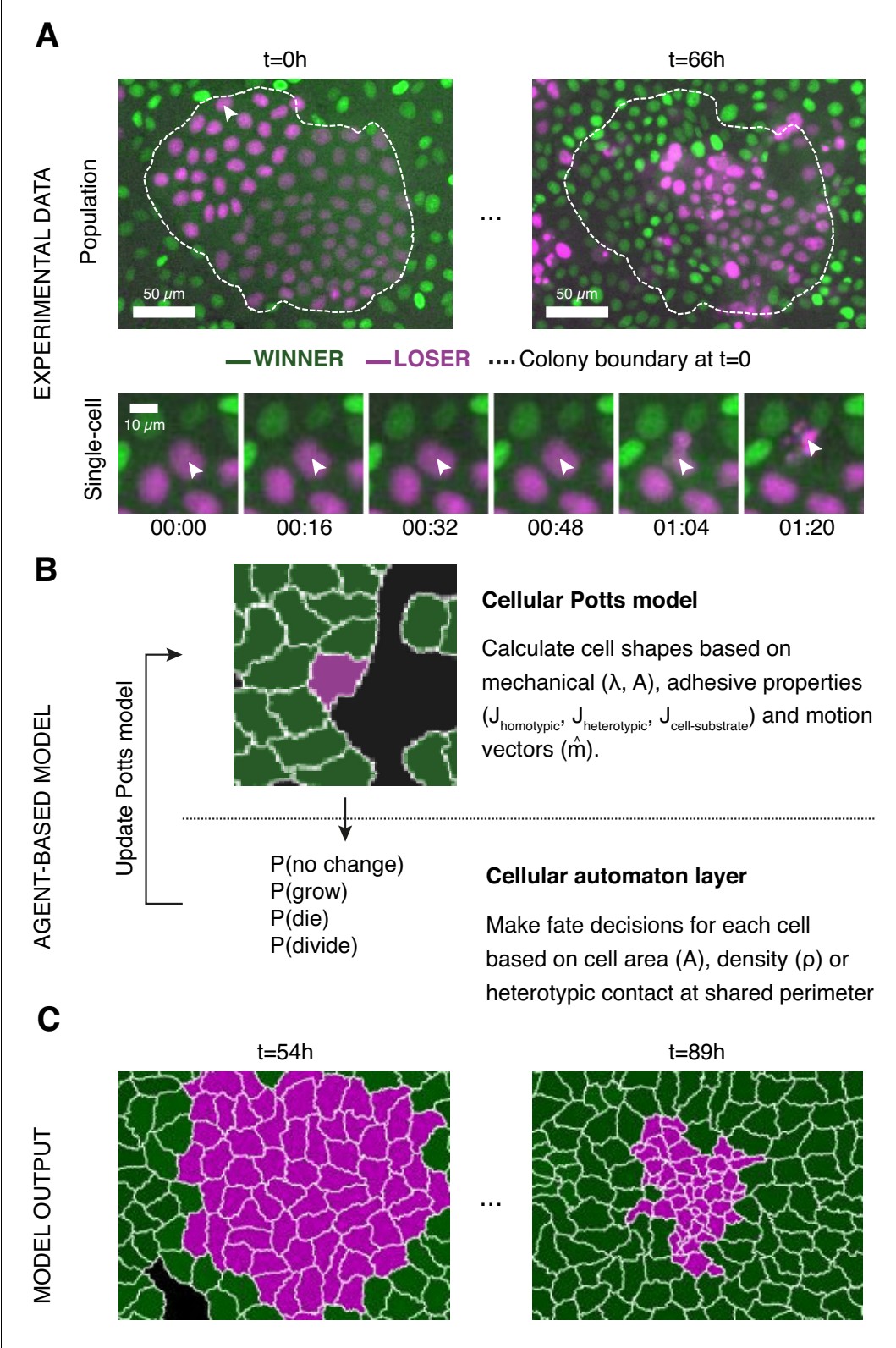

**Figure 1.** Experiments and simulations of cell competition. (**A**) Experimental snapshots of competition between MDCK[WT] cells (winner, green) and MDCK[Scrib] cells (loser, magenta) at the population scale (top) and the single-cell scale (bottom). Over the course of the experiment, the winner cells outcompete the loser cells whose area and number decreases (top) through apoptosis of individual cells (bottom). The arrowhead indicates the position of the dying cell examined in the bottom snapshots. The dashed line indicates the extent of the loser cell colony at the beginning of

*Figure 1 continued on next page*

*Figure 1 continued*

competition. (**B**) Framework of the multi-scale agent-based model used to simulate cell competition. The model consists of a Potts model and a cell automaton acting sequentially. The cellular Potts model first determines the cell shapes and position based on mechanical properties, such as intercellular adhesion and cell compressibility. Then a cell automaton decides whether cells grow, divide, or die based on a set of probabilistic rules that are algorithmically executed for each cell at each time point. These decisions are used to update the physical and geometrical properties of each cell before running the Potts model again. Further details about the cell behaviours included, the parameters, and the variables can be found in *Figure 1—figure supplements 1* and *2*. (**C**) The model outputs the time evolution of the organisation of the winner and loser cell types. These outputs can be quantitatively compared to experimental data.

The online version of this article includes the following figure supplement(s) for figure 1:

**Figure supplement 1.** Physical and decision-making components of the multi-layered computational model.

**Figure supplement 2.** Simulation workflow.

**Figure supplement 3.** Computational implementation of the adder model of growth followed by MDCK^WT cells.

**Figure supplement 4.** Density-dependent apoptosis.

**Figure supplement 5.** Contact-dependent apoptosis for biochemical competition.

## A multi-scale agent-based model for cell competition

To understand the emergence of cell competition, we implemented a multi-scale agent-based model that simulates mechanical interactions between cells and with their underlying substrate (*Figure 1B*, *Figure 1—figure supplement 1*, grey shaded area) and implements cell-autonomous decisions for growth, mitosis, and apoptosis (*Figure 1B*, *Figure 1—figure supplement 1*, pink shaded area). In contrast to existing computational approaches (*Rejniak and Anderson, 2011*; *Zhang et al., 2009*), our model includes the coupling between cellular mechanics and decision-making.

In our simulation, epithelial cells are modelled using a cellular Potts model (CPM) (*Graner and Glazier, 1992*), which enables physical interactions at the cell-cell and cell-substrate interfaces to be simulated (*Figure 1—figure supplements 1* and *2*, Materials and methods). This implementation was preferred to the less computationally costly vertex model (*Fletcher et al., 2014*) because we compare our model to our in vitro competition experiments (*Bove et al., 2017*) that start from a sub-confluent state.

In the CPM, each cell $\sigma^k$ possesses a cell type $\tau$ (winner or loser) and is represented by a set of pixels $(i,j)$. The free energy of a group of $N$ cells $\sigma^k$ is given by the Hamiltonian $H$:

$$H = \sum_{<i,j>} J\left(\tau\left(\sigma_{ij}^k\right), \tau\left(\sigma_{i'j'}\right)\right)\left(1 - \delta\left(\sigma_{ij}^k, \sigma_{i'j'}\right)\right) + \lambda \sum_{\sigma}\left(A\left(\sigma^k\right) - A_T\left(\sigma^k\right)\right)^2 \Theta(\tau) + \lambda_m \sum_{\sigma} \hat{m}(\sigma, t) \cdot \hat{s}.$$

The energy function represents energetic contributions due to intercellular adhesion, cell adhesion to the extracellular matrix, cell elasticity, and active cell movement. The first term in the Hamiltonian describes the adhesive interactions between cells at their shared interface, where δ represents the Kronecker delta function and Θ is the Heaviside theta function (see Materials and methods). When a cell interacts with other cells, it engages in either homotypic adhesion if they are of the same cell type or heterotypic adhesion if they are not. The respective adhesion energies are given by $J_{homotypic}$ and $J_{heterotypic}$. From a biological perspective, $J$ is the difference between the surface tension and intercellular adhesion. Therefore, higher $J$ implies lower intercellular adhesion. Cells can also interact with the substrate at their periphery via integrin binding to the extracellular matrix, parametrised by an adhesion energy $J_{cell-substrate}$. The second term describes an elastic energy with an elastic modulus $\lambda$ arising from the cytoskeleton. This energy scales with the difference between a cell's actual area $A(\sigma^k)$ and its target area $A_T(\sigma^k)$, which it would occupy in the absence of crowding due to other cells. The actual area is determined by the height and volume of a cell. The third term reflects energy due to cell motion and is parametrised by a kinetic energy $\lambda_m$ and a unit polarity vector $m(\sigma, t)$ that defines the direction of cell motion and undergoes rotational diffusion. An empirical conversion between computational time and experimental time was obtained by comparing the mean square displacement of isolated cells in experiments and simulations.

In addition to the CPM that determines cell shapes based on mechanical equilibrium, a second computational layer based on cell automata rules regulates changes in cell size due to growth and implements changes in cell fate (division and apoptosis) (*Figure 1B*, *Figure 1—figure supplement*

*1*, pink shaded area). It is in this layer that cellular decision-making is implemented at each time point based on a set of probabilistic rules that we determine from our experimental data (Materials and methods). We now briefly describe these rules and the calibration of their associated parameters (*Figure 1—figure supplement 2A*).

Experimental work has shown that MDCK cells maintain cell size homeostasis by following an 'adder' mechanism, in which each cell cycle adds a set volume to the cell (*Cadart et al., 2018*; *Figure 1—figure supplement 1B*). Following the adder model, we increase each cell's target area $A_T(t)$ at each time point $t$ and cells divide when a threshold area $\Delta A_{tot}$ has been added since the start of their cell cycle (*Figure 1—figure supplement 3*). The target area of cells at birth $A_T(0)$, the threshold area added at each cell cycle $\Delta A_{tot}$, and the maximum growth rate $G$ were all calibrated from movies of isolated cells to reflect the cell cycle time and size distribution measured in experiments (*Figure 1—figure supplement 2A*). Above a certain cell density, proliferation ceases – a phenomenon known as contact inhibition of proliferation (*Abercrombie, 1979*; *Lieberman and Glaser, 1981*). Arrest in proliferation is accompanied by a decrease in protein synthesis due to a drop in ribosome assembly and downregulation of the synthesis of cyclins (*Azar et al., 2010*). We incorporated contact inhibition of proliferation (*Figure 1—figure supplement 1E*) by making the target area growth rate $dA_T(t)/dt$ dependent on the difference between the actual cell area $A(t)$ and the target area $A_T(t)$ as $dA_T(t)/dt = Ge^{-k(A(t)-A_T(t))^2}$, where $G$ is the growth rate in the absence of crowding and $k$ is a heuristic parameter that quantifies the sensitivity to contact inhibition. As a result, cellular growth rate slows down exponentially as crowding increases, leading to an increase in cell cycle time.

We implemented two separate rules for cell apoptosis in mechanical and biochemical competition (*Figure 1—figure supplement 1C*). First, under crowded conditions where mechanical competition is dominant, the probability of apoptosis $p_{apo}$ increases with local cell density $\rho$ following a sigmoid curve (*Figure 1—figure supplement 4A*). Our experimental data suggested that winner cell apoptosis followed the same law as losers except shifted towards higher local densities (*Bove et al., 2017*). Second, in biochemical competition, experimental data indicates that the probability of apoptosis of loser cells depends on the percentage of their perimeter in contact with the winner cells (heterotypic contact, *Figure 1—figure supplement 5A*; *Levayer et al., 2015*). This was implemented as a Hill function as a function of percentage of perimeter occupied by heterotypic contact (Materials and methods). In the absence of experimental measurements, we chose a maximum probability $p_{apomax}$ of death per frame of a similar magnitude to that measured in mechanical competition. This is justified by the fact that mechanical and biochemical competition takes place over comparable durations in MDCK cells ~2–4 days (*Hogan et al., 2009*; *Kajita et al., 2010*).

In experiments, cell elimination can also occur through live cell extrusions when the cell apical area decreases substantially compared to the population average (*Eisenhoffer et al., 2012*; *Kocgozlu et al., 2016*). In our simulations, cells delaminated when their actual area $A(t)$ became smaller than $\frac{<A_i>}{2}$, with $<A_i>$ the average area of all cells in the simulation at time $t$ (*Figure 1—figure supplement 1D*).

Taken together, the combination of cellular mechanics and decision-making strategies provides a multi-scale agent-based model to investigate how the interplay between short-range and long-range competitive interactions determines tissue composition. Many parameters are used to describe each cell type's behaviour, some can be measured directly from experiments while others must be empirically determined based on comparison of the output of the simulations and the experimental data (*Figure 1—figure supplement 2A*, *Supplementary files 1* and *2*). We sought to fix as many parameters as possible to restrict the parameter space explored.

## Growth and homeostasis of pure cell populations

To validate the predictive power of our model, we first simulated homeostasis in pure cell populations of winner and loser cells undergoing proliferation and apoptosis. To calibrate our model parameters (*Figure 1—figure supplement 2A*), we compared simulations to experiments on the basis of the temporal evolution of population metrics, such as cell count and average cell density (*Figure 2C, D, F, G*). In addition, as experimental and theoretical work has shown that cell organisation in monolayers can be described by the distribution of number of neighbours each cell possesses (*Gibson et al., 2006*) and their area relative to the population mean (*Farhadifar et al., 2007*), we

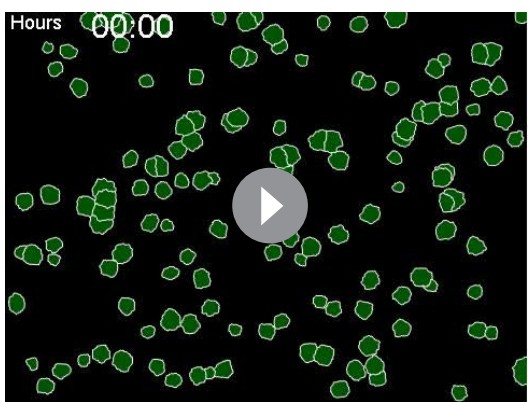

**Video 1.** Simulation of the growth of a pure winner cell epithelium (*Figure 2A*).
https://elifesciences.org/articles/61011#video1

also used these cell-scale metrics for confluent epithelia (*Figure 2E, H*, *Figure 2—figure supplement 1*).

At the start of our experiments, many isolated cells can be observed (t=0h, *Figure 2B*, *Figure 2—figure supplement 2A–C*). In pure populations, MDCK$^{Scrib}$ cells spread markedly more than MDCK$^{WT}$ (*Figure 2—figure supplement 2B, D*, compare MDCK$^{WT}$ in *Video 2* to MDCK$^{Scrib}$ in *Video 3*), consistent with *Wagstaff et al., 2016*. Interestingly, in mixed populations dominated by WT cells, MDCK$^{Scrib}$ cell area diminished compared to pure populations even prior to confluence (*Figure 2—figure supplement 2C–E*). We used these measurements to set the distribution in cell areas at birth $A_T(0)$ and the area added at each cell cycle $\Delta A_{tot}$ for each cell type in pure and competitive conditions. Transcriptomic data comparing both cell types indicates that MDCK$^{Scrib}$ do not express more integrins than MDCK$^{WT}$ (*Wagstaff et al., 2016*). Therefore, we assigned the same value of $J_{cell\text{-}substrate}$ to both cell types.

The maximum growth rate $G$ was parameterised based on the measured distribution of cell cycle durations prior to confluence (*Bove et al., 2017*), which showed that losers grew significantly slower than winners (mean cell cycle times: MDCK$^{Scrib}$ ~21.6 hr vs. MDCK$^{WT}$ ~18 hr). Therefore, we assigned a smaller $G$ to losers than to winners.

After confluence, cell shape is controlled by the interplay between intercellular adhesion energy $J_{homotypic}$ and the stiffness modulus $\lambda$. The value of $J_{homotypic}$ was adjusted such that, at confluence, the distributions in apical area, number of neighbours, and area relative to the population mean matched experiments (*Figure 2—figure supplement 1A, B*), as done by others (*Farhadifar et al., 2007*). An accurate replication of sidedness of cells is particularly important for simulating biochemical competition because the probability of apoptosis of loser cells is linked to the fraction of their perimeter contacting winner cells (*Levayer et al., 2015*). Previous experimental work showed that, in competition experiments, the area of loser cells became significantly smaller than that of winners when monolayers reached high densities post-confluence (*Wagstaff et al., 2016*; *Bove et al., 2017*). Therefore, we assigned losers a smaller value of $\lambda$ than winners.

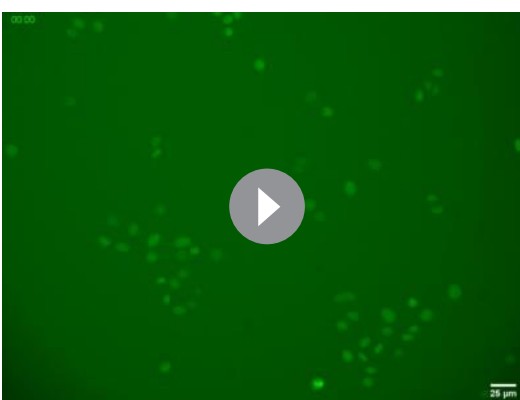

**Video 2.** Representative growth of a pure MDCK$^{WT}$ epithelium over 66 hr (*Figure 2B*). The nucleus of MDCK$^{WT}$ cells is labelled with H2B-GFP. Scale bar 25 µm.
https://elifesciences.org/articles/61011#video2

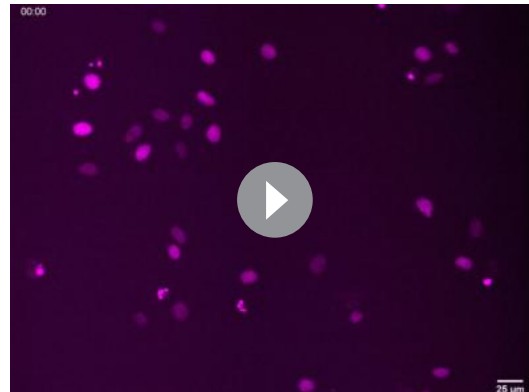

**Video 3.** Representative growth of a pure MDCK$^{Scrib}$ epithelium over 79 hr (*Figure 2F–H*). The nucleus of MDCK$^{Scrib}$ cells is labelled with H2B-mRFP. Scale bar 25 µm.
https://elifesciences.org/articles/61011#video3

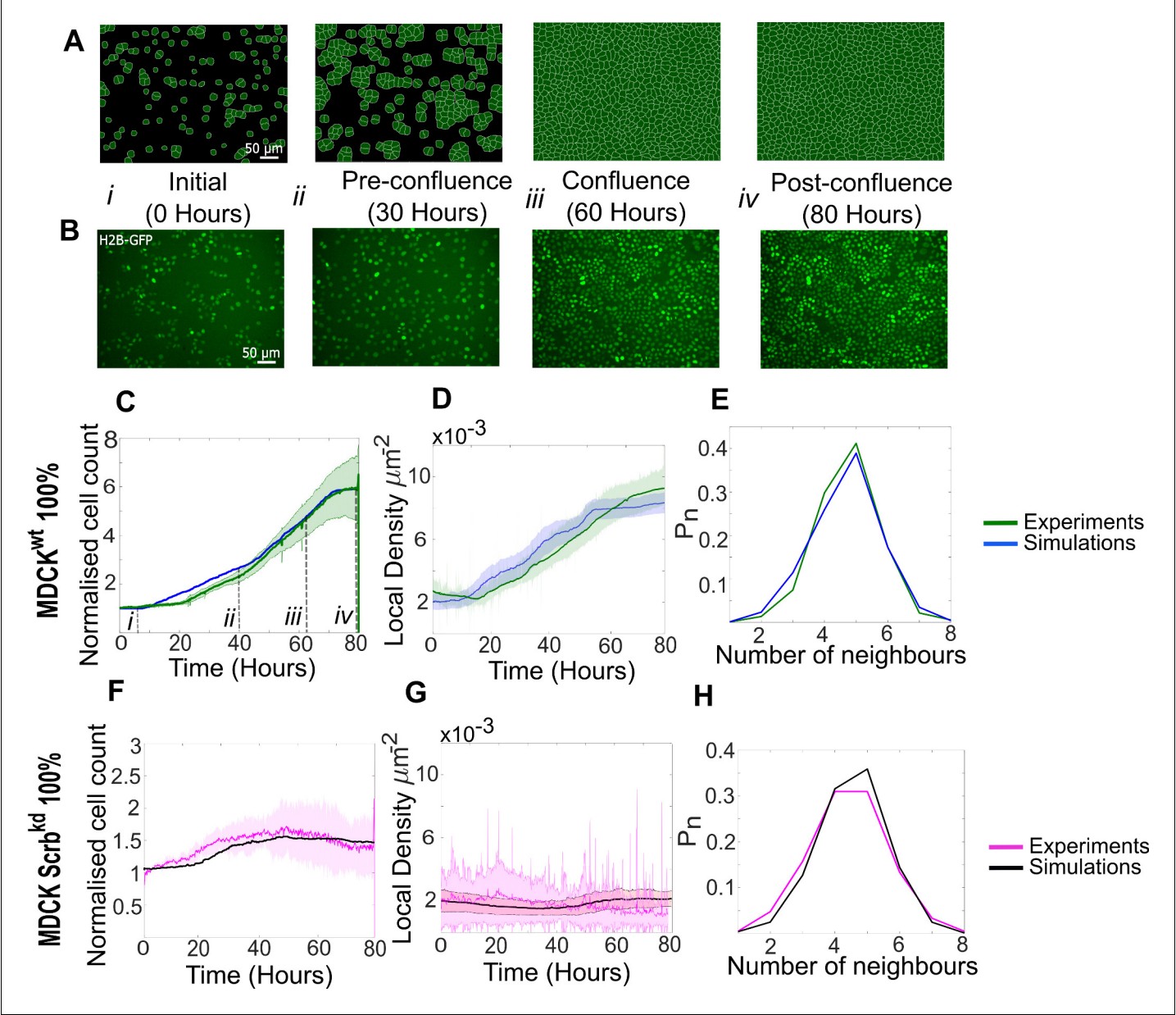

**Figure 2.** Model simulations capture the dynamics of growth and homeostasis in pure cell populations. (A) Simulation snapshots of the growth of a pure population of winner cells (*Video 1*). Cells are initially separated by free space (black). Each image corresponds to one computational field of view, representing 530 μm × 400 μm. (B) Experimental snapshots of MDCK^WT (winner) cells expressing the nuclear marker H2B-GFP (*Video 2*). Each image corresponds to 530 μm × 400 μm and is acquired by wide-field epifluorescence using ×20 magnification. The timing of each image is indicated between the two image rows. (C) Normalised cell count as a function of time for winner cells in simulations (blue) and MDCK^WT experiments (green). (i–iv) indicate the time points at which the snapshots in (A, B) were taken. (D) Average local cellular density as a function of time for the same data as (C). (E) Distribution of sidedness of cells post confluence. The curves indicate the proportion of cells as a function of number of neighbours. (C–E) Green curves represent experimental data and blue curves simulated data. (F–H) are same as (C–E) for pure populations of loser cells (MDCK^Scrib). Magenta curves are experimental data (*Video 3*), and black curves represent simulated data. (C–H) Data are pooled from three biological replicates imaging four fields of view each and from 12 simulations. The solid line indicates the mean, and the shaded area indicates the standard deviation. Parameters used for the simulations in this figure are listed in *Supplementary file 1*.

The online version of this article includes the following figure supplement(s) for figure 2:

**Figure supplement 1.** Comparison of epithelium organisation in simulations and experiments for pure populations.

**Figure supplement 2.** Cell area in pure and mixed populations prior to confluence.

The strength of contact inhibition, $k$ (Materials and methods), was adjusted empirically such that the average local cell density at long time scales in simulations reached a plateau that matched the one observed in experiments (*Figure 2D, G*).

In mechanical competition experiments, loser cells undergo apoptosis when they are in crowded environments. Therefore, we implemented a relationship between probability of apoptosis per cell per unit time ($p_{apo}$) and the local cell density $\rho$ parametrised by fitting our experimental data with a sigmoid function (Materials and methods, *Figure 1—figure supplement 4A*; *Bove et al., 2017*). The local cell density $\rho$ was defined as the inverse of the sum of the area of the cell of interest and its first neighbours (Materials and methods, *Figure 1—figure supplement 4B*). While $p_{apo}$ saturates at high densities for loser cells, experimental data was not available for the highest densities for winner cells (data points, *Figure 1—figure supplement 4A*). Therefore, we assumed that the maximum probability of apoptosis $p_{apo\ max}$ was the same for winner and loser cells (*Figure 1—figure supplement 4A*).

When the simulations were initialised with the calibrated parameters and the same initial cell number as in experiments, cell count and density in the simulations qualitatively reproduced our experimental observations (*Figure 2A, B*, *Videos 1* and *2*). MDCK$^{WT}$ cell count increased for ~70 hr before reaching a plateau at a normalised cell count of 5.5, indicative of homeostasis (*Figure 2C*). The temporal evolution of the average local cell density and the distribution of number of neighbours at confluence were also faithfully replicated by our simulations (*Figure 2D, E*). Similarly, our parametrisation of MDCK$^{Scrib}$ accurately replicated the temporal evolution of cell count and density, as well as the distribution of the number of cell neighbours (*Figure 2F–H*, *Video 3*). In particular, the loser cell count and density stayed fairly constant throughout the whole simulation and experiment.

## Model epithelia maintain a homeostatic density

The maintenance of an intact barrier between the internal and the external environment is a key function of epithelia. This necessitates exact balancing of the number of cell deaths and divisions. Failure to do so results in hyperplasia, an early marker of cancer development. Previous work has revealed that epithelia possess a preferred density to which they return following perturbation, signifying that they seek to maintain a homeostatic density (*Eisenhoffer et al., 2012*; *Marinari et al., 2012*; *Gudipaty et al., 2017*). In the experiments performed in *Eisenhoffer et al., 2012*, cells were grown to confluence on stretchable substrates and subjected to a step deformation in one axis. When deformation increased cellular apical area, the frequency of cell division increased (*Gudipaty et al., 2017*), while a decrease in apical area resulted in increased live cell extrusion and apoptosis (*Eisenhoffer et al., 2012*). Therefore, the existence of a homeostatic density is an essential property of epithelia that relates to their sensitivity to crowding – a key factor in mechanical competition. However, current models of epithelia do not implement this.

In our model, we implemented two mechanisms shown experimentally to decrease cell density: cell extrusions and density-dependent apoptoses (*Figure 1—figure supplement 1C, D*, Materials and methods). We simulated the response of a confluent epithelium to a sudden 30% increase in homeostatic density. In experiments on confluent MDCK$^{WT}$ epithelia (*Eisenhoffer et al., 2012*), a sudden increase in crowding was followed by a gradual decrease in cell density resulting from a combination of apoptoses and live cell extrusion, before returning to the initial homeostatic density after ~6 hr (green data points, *Figure 3A*).

In our simulations, we allowed MDCK$^{WT}$ cells to reach their homeostatic density before suddenly increasing the cell density by a percentage similar to experiments. In response to this, cell density decreased gradually over a period of 6 hr with dynamics similar to those determined experimentally (solid black line, *Figure 3B*). Thereafter, cell density remained at the homeostatic density for long time periods (*Figure 3B*). Thus, our model implementation and parametrisation replicate the return to homeostasis of pure populations of MDCK$^{WT}$ cells. As the relationship between $p_{apo}$ and $\rho$ was fitted using the data from our competition experiments, this also raises the possibility that cell apoptosis in response to crowding is a cell-autonomous process rather than specific to cell competition.

## Density-mediated apoptosis is sufficient to explain experimental observations in mechanical competition

Our experimental work indicated that, in competitions between MDCK$^{WT}$ and MDCK$^{Scrib}$, two processes might be at play, a density-dependent apoptosis of MDCK$^{Scrib}$ and an upregulation of

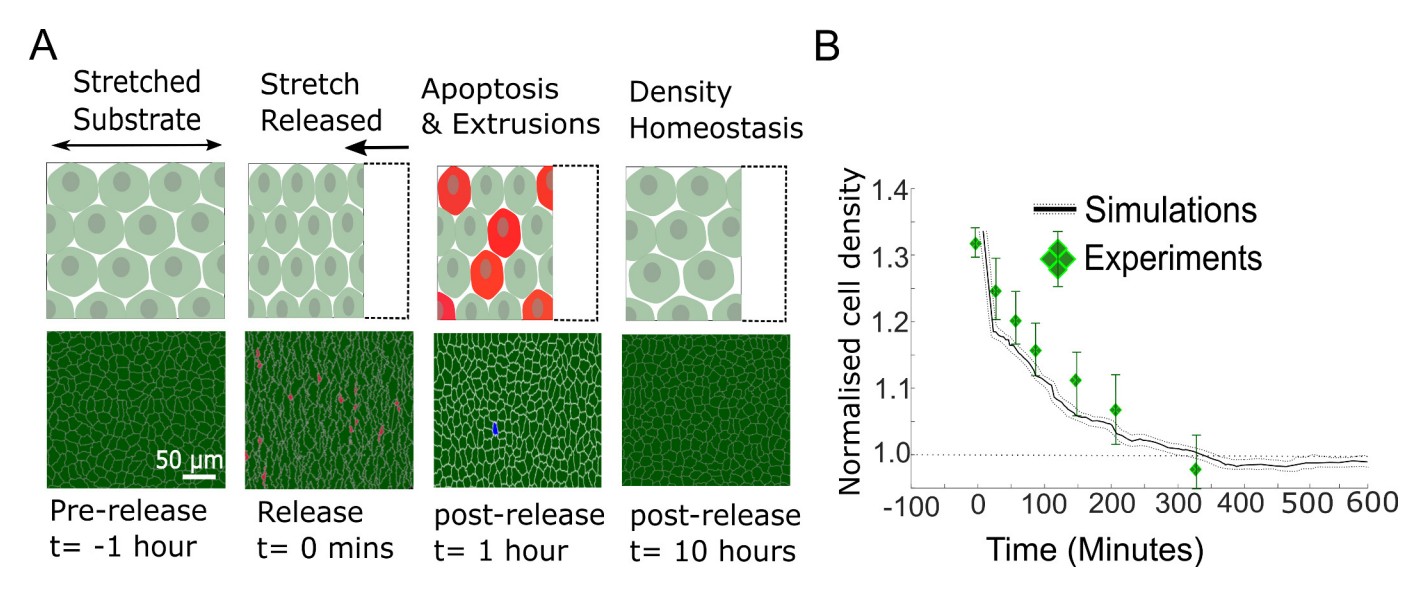

**Figure 3.** Model epithelia return to homeostasis in response to a sudden increase in local density. (**A**) Winner cells grown to confluence on a stretched silicon substrate are subjected to a sudden increase in density after stretch release. The monolayer returns to its homeostatic density over time through extrusions and density-mediated apoptoses. Top row: cartoon diagrams depicting the experiment. Bottom row: snapshots of simulations. Cells eliminated by live cell extrusion are shown in red and by apoptosis in blue. (**B**) Evolution of cell density as a function of time in response to a step increase in cell density at t = 0 min. Density in simulations is indicated by the black line, and experimental data from *Eisenhoffer et al., 2012* are shown by green diamond markers. The shaded region around the black line indicates the standard deviation of n = 5 simulations. The whiskers around the diamond markers indicate the standard deviation of the experimental measurements. The dashed horizontal line denotes the initial cell density.

division of MDCK$^{WT}$ in MDCK$^{Scrib}$-dominated neighbourhoods (*Bove et al., 2017*). The former is central to mechanical competition, while the latter implies that contact between cell types may control division rate. To determine which process was dominant, we tested whether cell-type differences in density-mediated apoptosis alone were sufficient to explain competition.

We used our model of winner and loser cells with different sensitivities to crowding parametrised from experiments on pure cell populations (*Figure 2*, *Supplementary file 1*). We initialised our simulations by seeding a 90:10 winner-to-loser cell ratio, as in experiments. Our simulations were able to quantitatively reproduce the experimental data for competition dynamics, with no further adjustment in parameters. As in the experiments, simulated winner cells (green) rapidly proliferated while loser cell numbers (red) increased weakly until ~50 hr before diminishing (*Figure 4A, B*, *Videos 4* and *5*). Furthermore, the evolution of cell count was quantitatively replicated over the entire duration of the experiment for both winner and loser cells (*Figure 4C*). Cumulative divisions and apoptoses in simulations closely matched those observed in experiments (*Bove et al., 2017*; *Figure 4—figure supplement 1B, C*). One of the most striking features of experimental data is that the local density of loser cells in competition increases dramatically compared to pure populations (approximately fivefold increase, *Figure 4—figure supplement 1D* for comparison), while the local density of winner cells in competition follows the same trend as in pure populations (*Figure 4—figure supplement 1E* for comparison) (*Bove et al., 2017*). The sharp increase in local density of loser cells is replicated in our simulations (red curve, *Figure 4D*) and likely arises from their lower stiffness modulus $\lambda$. In addition, the probability of apoptosis and division as a function of density computed from simulation data matched the experimentally measured ones for both cell types (*Figure 4E, F*). While the former is an input to our simulation, the latter is an output. Finally, when we compared the probability of division of winner cells in contact with at least one loser cell to that of winner cells in contact with only winner cells, we found that the probability of division of winner cells increased when they were in contact with loser cells (*Figure 4G*). This was consistent with our experimental observations and occurred despite the fact that we did not implement any aspect of biochemical signalling in our simulations, signifying it represents an emergent property.

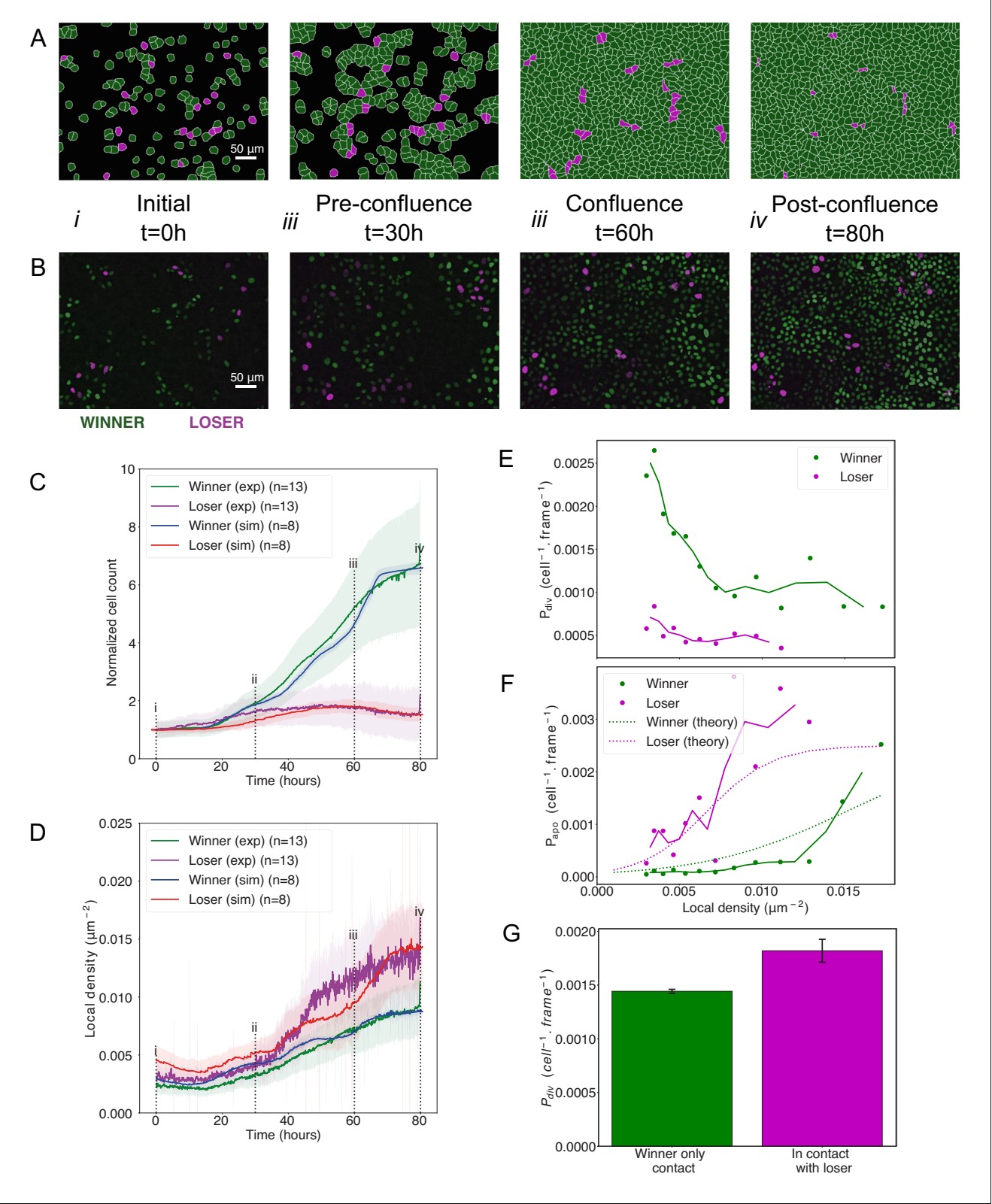

**Figure 4.** Cell competition in co-cultures of cells with different homeostatic densities. (**A**) Simulation snapshots of competition between 90% winner (green) and 10% loser cells (red) (***Video 4***). Cells are initially separated by free space (black). Each image corresponds to 530 µm × 400 µm. (**B**) Experimental snapshots of competition between 90% MDCK^WT cells (winner, green) and 10% MDCK^Scrib cells (loser, red) (***Video 5***). MDCK^WT express the nuclear marker H2B-GFP, while MDCK^Scrib express the nuclear marker H2B-RFP. Each image corresponds to 530 µm × 400 µm and is acquired by

*Figure 4 continued on next page*

*Figure 4 continued*

wide-field epifluorescence using ×20 magnification. The timing of snapshots is indicated in between rows A and B. Scale bars represent 50 µm. **(C)** Temporal evolution of normalised cell count for winner cells (simulations: blue line; experiments: green line) and loser cells (simulations: red line; experiments: purple line) in experiments and simulations initiated with a 90:10 ratio of winner:loser cells. Data are pooled from three biological replicates imaging four fields of view for the experiments and from 12 simulations. **(i–iv)** indicate the time points at which the snapshots in (A, B) were taken. **(D)** Temporal evolution of the local cell density for the simulations and experiments shown in (C). The local density is defined in *Figure 1—figure supplement 4B*. **(C, D)** Solid lines indicate the average of the data, and the shaded area indicates the standard deviation. **(E)** Probability of division per cell per frame as a function of local density predicted from simulations. Markers indicate probability of division for each density bin, and solid lines indicate moving average. **(F)** Probability of apoptosis per cell per frame as a function of local density predicted from simulations compared to the theoretical input functions implemented in the model. Markers indicate probability of apoptosis for each density bin, and solid lines indicate moving average. Dashed lines show the input functions implemented in the model based from experimental data in *Bove et al., 2017*; *Figure 1—figure supplement 4A*. **(G)** Probability of division per cell per frame for winner cells in contact with winner only (green bar) and in contact with at least one loser cell (red bar). Whiskers indicate the coefficient of variation $cv$ calculated for each. The number of cells observed $N$ and number of divisions $n$ were respectively $N = 4.3 \times 10^6$ and $n = 6246$ for winner contact only and $N = 1.6 \, 10^5$ and $n = 292$ for winners in contact with losers. Data was gathered from 10 simulations. Parameters used for the simulations in this figure are in *Supplementary file 1*.

The online version of this article includes the following figure supplement(s) for figure 4:

**Figure supplement 1.** Comparison between simulations and experiments for mechanical competition.

---

Overall, differences in density-dependent apoptosis alone are sufficient to replicate the evolution of cell count and density observed in competition between MDCK$^{WT}$ and MDCK$^{Scrib}$ as well as the upregulation of MDCK$^{WT}$ division in MDCK$^{Scrib}$-dominated neighbourhoods (*Bove et al., 2017*), suggesting that mechanical competition represents the dominant mechanism of population change in these experiments.

## Differences in homeostatic density and cell stiffness control the outcome of mechanical cell competition

To understand the mechanistic origin of density-mediated cell competition, we varied the growth rate $G$, the intercellular adhesion $J_{heterotypic}$, the stiffness $\lambda$, and the contact inhibition parameter $k$ for the individual cell types starting from an initial set of values that gave rise to mechanical competition (*Supplementary file 2*). We reasoned that, in a competition setting, the values of each parameter in one cell type relative to the other were likely more important than their absolute values. Therefore, we varied each parameter in only one of the two cell types.

We first varied the growth rate $G$ of loser cells (*Supplementary file 2*). $G$ controlled the time required for elimination and the peak loser cell count but did not affect the outcome of competition with losers being eliminated for all growth rates examined (*Figure 5—figure supplement 1A, B*), consistent with simulations of population dynamics based on ordinary differential equations (*Basan et al., 2009*).

Second, we varied the heterotypic adhesion energy, $J_{heterotypic}$, between the winners and the losers between ±50% of the value of the homotypic adhesion $J_{homotypic}$ (*Supplementary file 2*). When $J_{heterotypic}$ is larger than $J_{homotypic}$, cells preferentially adhere to cells of their own type. In our simulations, $J_{heterotypic}$ did not appear to change the kinetics or the outcome of competition (*Figure 5—figure supplement 1C*). Note that varying $J_{homotypic}$ in one of the cell types only would have similar effects to a variation in $J_{heterotypic}$.

In our simulations, sensitivity to contact inhibition $k$ was chosen to be the same for both cell types. This parameter constrains how far cells can

**Video 4.** Simulation of mechanical competition between 90% winner cell types and 10% loser cell types for default parameters (*Figure 4A, C, D*). The different shades of green represent the different generations of winner cells, and the different shades of red represent the different generations of loser cells.
https://elifesciences.org/articles/61011#video4

deviate from their target area $A_T$ before their growth rate $G(t)$ approaches 0 and they stop growing (Materials and methods, *Supplementary file 2*). In pure winner cell populations, the average local density reached a plateau after confluence defining a homeostatic density (*HD*), which decreased with increasing contact inhibition $k$ (*Figure 5A*, blue line). However, this effect was not observed in pure loser populations (*Figure 5A*, red line) because their probability of apoptosis is high even for densities below the homeostatic density dictated by $k$ (*Figure 1—figure supplement 4A*). Indeed, under normal growth conditions, we predict that loser cells never reach densities where contact inhibition parametrised by $k$ becomes active. In all cases, the homeostatic density of winner cells was higher than in loser cells but the difference in homeostatic density, $\Delta HD$, decreased with increasing $k$ (*Figure 5A*). Thus, in winner cells, homeostatic density is controlled by a decrease in growth controlled by the contact inhibition parameter $k$, while in loser cells it is controlled by density-dependent apoptosis (*Figure 1—figure supplement 4A*).

In competitions, when we varied the homeostatic density of the winner cells (by changing $k$, *Supplementary file 2*), we found that, after 80 hr, loser cells were completely eliminated for high values of $\Delta HD$ but they survived when $\Delta HD$ was lower (*Figure 5B, C, E*). In addition, the time required for elimination of 50% of loser cells increased with decreasing $\Delta HD$ and increasing $k$ (*Figure 5A, F*). Interestingly, loser cell count appeared to converge towards a non-zero plateau for values of $k$ larger than 0.1 at long time scales (*Figure 5—figure supplement 1D*). Therefore, the difference in homeostatic density, $\Delta HD$, between the winner and loser cells governs the kinetics and the outcome of mechanical competition for the durations examined in this study (*Figure 5A, C ,F*, *Figure 5—figure supplement 1D*).

As in our initial parameterisation the winner cells have a higher stiffness $\lambda$, the loser cells are compressed by the winners during competition. As a result, the average local density of loser cells is larger than that of winners, which, combined with their greater sensitivity to crowding, leads to increased apoptosis. To determine the impact of $\lambda$ on competition, we varied the loser cell stiffness while maintaining winner cell stiffness constant. When the relative stiffness parameter $\Lambda = \lambda_{loser}/\lambda_{winner}$ was smaller than 1, loser cells were eliminated (*Figure 5D*, *Figure 5—figure supplement 1E, F*). By contrast, when $\Lambda$ was equal to or higher than 1, loser cells survived (*Figure 5D*, *Figure 5—figure supplement 1E, G*). Akin to $\Delta HD$, changes in the ratio of winner-to-loser cell stiffness altered the kinetics of competition and its outcome over durations considered in this study (*Figure 5E*, *Figure 5—figure supplement 1E*). These results are consistent with experiments showing that competition is decreased in the presence of an inhibitor of Rho-kinase (Y27632) that reduces cell contractility in both populations (*Wagstaff et al., 2016*).

When $\Delta HD$ was low or $\Lambda$ was larger than 1, the change in competition outcome occurred because of a decrease in the local density of loser cells in mixed populations, which in turn led to decreased apoptosis. However, winners have an extra competitive edge because when free space becomes available due to cell death or cell area compressibility, they take advantage of the free space due to faster growth using a *squeeze and take* or a *kill and take* tactic (*Gradeci et al., 2020*). At very long times, this effect alone may be sufficient for them to dominate in mixed populations.

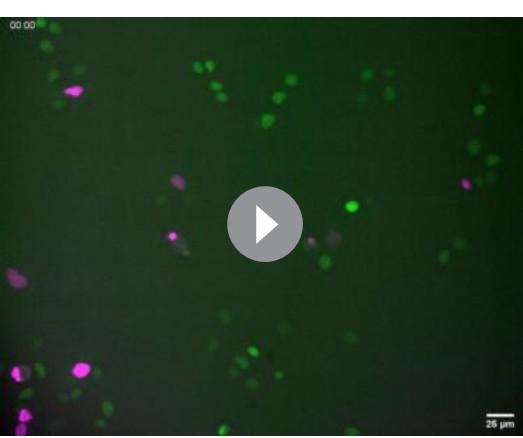

**Video 5.** Representative competition between 90% MDCK[WT] and 10% MDCK[Scrib] over 66 hr (*Figure 4B–D*). MDCK[WT] nuclei are marked with H2B-GFP (green), and MDCK[Scrib] nuclei are marked with H2B-mRFP (magenta). Scale bar 25 µm.
https://elifesciences.org/articles/61011#video5

## Tissue organisation predicts the kinetics of biochemical competition

Our simulation can also be used to gain mechanistic insights into biochemical competition. Recent work has shown that, during biochemical competition, apoptosis in loser cells is governed by the extent of their contact with winner cells (*Levayer et al., 2016*) and that perturbations that increase mixing between cell types increase competition (*Levayer et al., 2015*).

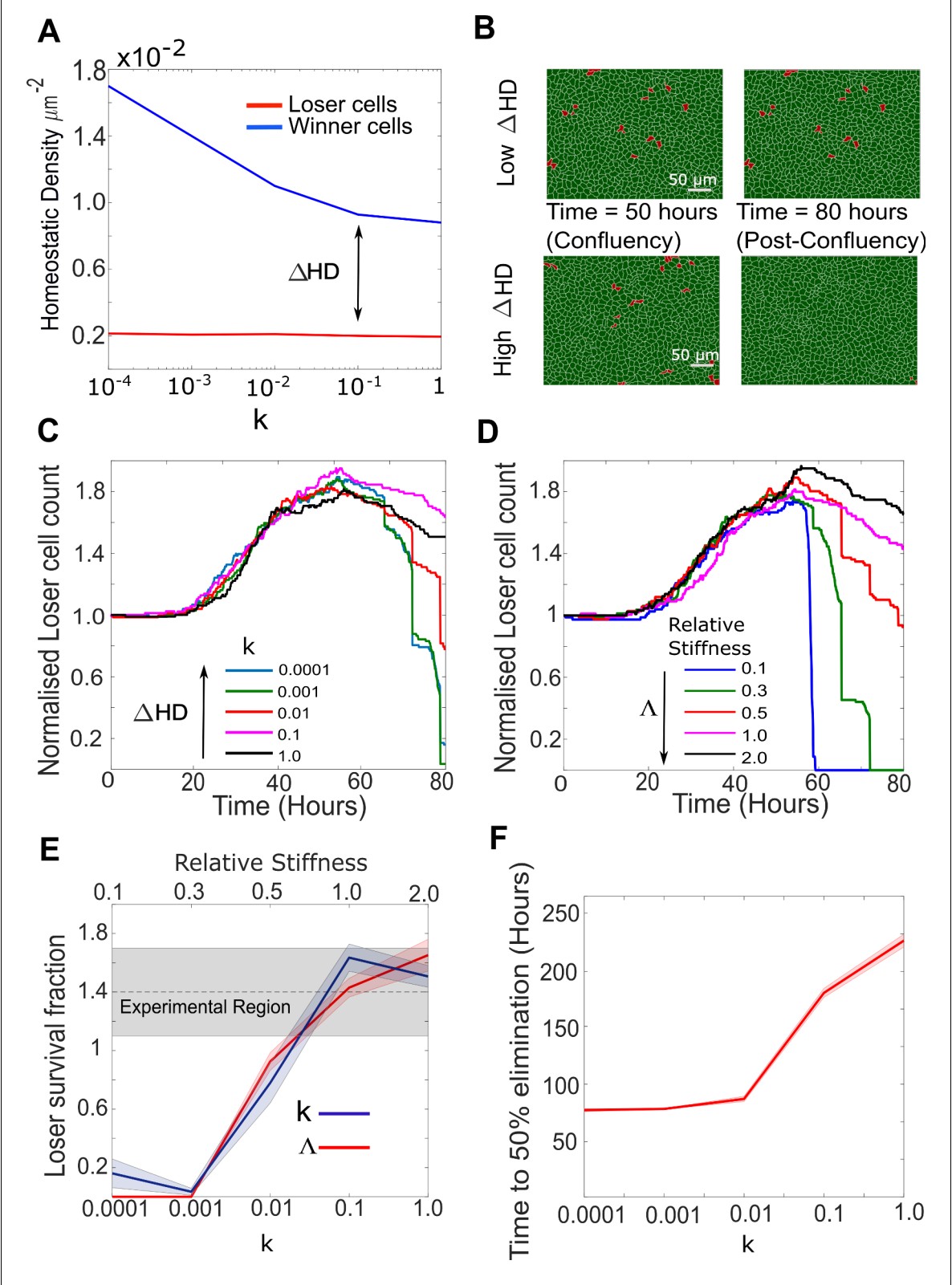

**Figure 5.** Cellular stiffness and homeostatic density control the outcome of mechanical competition. (**A**) Evolution of homeostatic density as a function of the parameter *k*, quantifying the sensitivity to contact inhibition. Data are shown for pure populations of winner (blue) and loser (red) cells. (**B**) Simulation snapshots of competition between 90% winner cells (green) and 10% loser cells (red). The top two panels show the population evolution for a small difference in homeostatic density Δ*HD* between the two cell types. The bottom two panels show the population evolution for a large Δ*HD*

*Figure 5 continued on next page*

*Figure 5 continued*

between the two cell types. Winner cells are shown in green and loser cells in red. Each image corresponds to 530 μm × 400 μm. (**C**) Normalised cell count for loser cells in competition simulations for different values of the contact inhibition parameter $k$ in the winner cells. As $k$ decreases, $\Delta HD$ increases. Temporal evolution of cell count in longer simulations is shown in *Figure 5—figure supplement 1D*. (**D**) Normalised cell count for loser cells in competition simulations for different values of the relative stiffness parameter $\Lambda = \lambda_{loser}/\lambda_{winner}$. Winner cells have a fixed stiffness of 1.0. Temporal evolution of cell count in longer simulations is shown in *Figure 5—figure supplement 1E*. Snapshots of competition are shown in *Figure 5—figure supplement 1F, G* for $\Lambda = 0.3$ and $\Lambda = 2$. (**E**) Loser cell survival fraction after 80 hr in simulations run with different parameters for relative stiffness $\Lambda$ (red line) and contact inhibition $k$ (blue line). Shaded blue and red regions denote the standard deviation. The grey shaded region indicates the survival fraction of loser cells observed in experiments after 80 hr. The x-axis scale is indicated on the top of the graph for $\Lambda$ and on the bottom of the graph for $k$. (**F**) Time to 50% elimination of loser cells as a function of the contact inhibition parameter. The solid line indicates the mean and the shaded regions the standard deviation. Parameters used for the simulations in this figure are in *Supplementary file 2*.

The online version of this article includes the following figure supplement(s) for figure 5:

**Figure supplement 1.** Effect of simulation parameters on mechanical competition.

To study biochemical competition in isolation from any mechanical effect, we assumed that both cell types have identical stiffnesses $\lambda$, equal sensitivities to contact inhibition $k$, and high but equal homeostatic densities (*Supplementary file 3*). In both cell types, we modelled the dependency of apoptosis on the proportion of cell perimeter $p$ engaged in heterotypic contact by using a Hill function parameterised by a steepness $S$ and an amplitude $p_{apo\,max}$:

$$p_{apo}(p) = \frac{p_{apo\,max}\,p^n}{(S^n + p^n)},$$ where $n$ is the Hill coefficient (*Figure 1—figure supplement 5A*).

When $S$ decreases, the probability of apoptosis increases rapidly with the extent of heterotypic contact (*Figure 1—figure supplement 5A*). For winner cells, we chose a low $p_{apo\,max}$ and high $S$ because we do not expect their apoptosis to show sensitivity to contact with loser cells. In contrast, for loser cells, we chose $p_{apo\,max}$ to be 10-fold higher than in winners, giving an amplitude similar to the maximal probability of apoptosis observed in losers during mechanical competition and similar kinetics of elimination, as observed in experiments (*Hogan et al., 2009*; *Figure 1—figure supplement 4A*, *Figure 1—figure supplement 5A, B*). To investigate biochemical competition, we varied parameters modulating contact between cells (the heterotypic adhesion $J_{heterotypic}$), apoptosis of losers (the Hill function parameter $S$ of loser cells), as well as tissue organisation.

First, we assumed a homogenous seeding of each cell type with a 50:50 ratio between winners and losers. Competition depends on the relative probability of apoptosis $p_{apo}$ in winners and losers as a function of the fraction of their perimeter $p$ in heterotypic contact (*Figure 1—figure supplement 1C*, right). Therefore, varying $p_{apo\,max}$ or $S$ has a qualitatively similar overall effect on competition. We examined the dependency of competition outcome on $S_{Loser}$ with $S_{Winner}$, $p_{apo\,max,\,winner}$, and $p_{apo\,max,\,loser}$ fixed (*Supplementary file 3*). For all values of $S_{Loser}$, loser cells first increased in number until overall confluence at ~60 hr, before decreasing after that (*Figure 1—figure supplement 5B*). When $S_{Loser}$ was low, losers were eliminated because $p_{apo}$ for losers was higher than for winners for all heterotypic contact extents (*Figure 1—figure supplement 5B, C*, *Video 6*), whereas when $S_{Loser}$ was high, winners and losers had comparable $p_{apo}$ when in heterotypic contact, leading to coexistence because no competition took place (*Figure 1—figure supplement 5B, C*).

Next, we investigated the dependency of cell competition on initial seeding conditions for values of $S_{Loser}$ that gave rise to competition ($S_{Loser} = 0.3$, $S_{Winner} = 0.5$, *Supplementary file 3*). We examined three different initial seeding conditions: fully mixed (*Figure 6A*, middle column), partially sorted with loser cells grouped

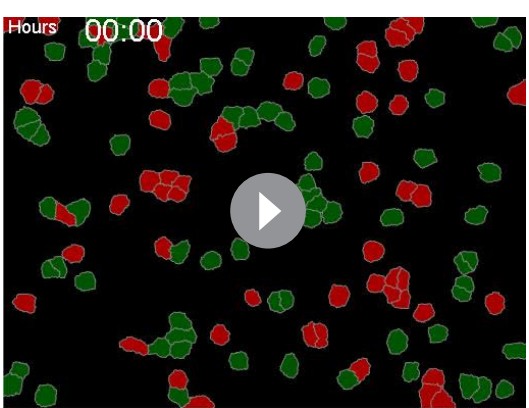

**Video 6.** Simulation of biochemical competition between 50% winner cell types and 50% loser cell types in an initial fully mixed configuration for a low value of the steepness $S = 0.1$ (*Figure 1—figure supplement 5C*).
https://elifesciences.org/articles/61011#video6

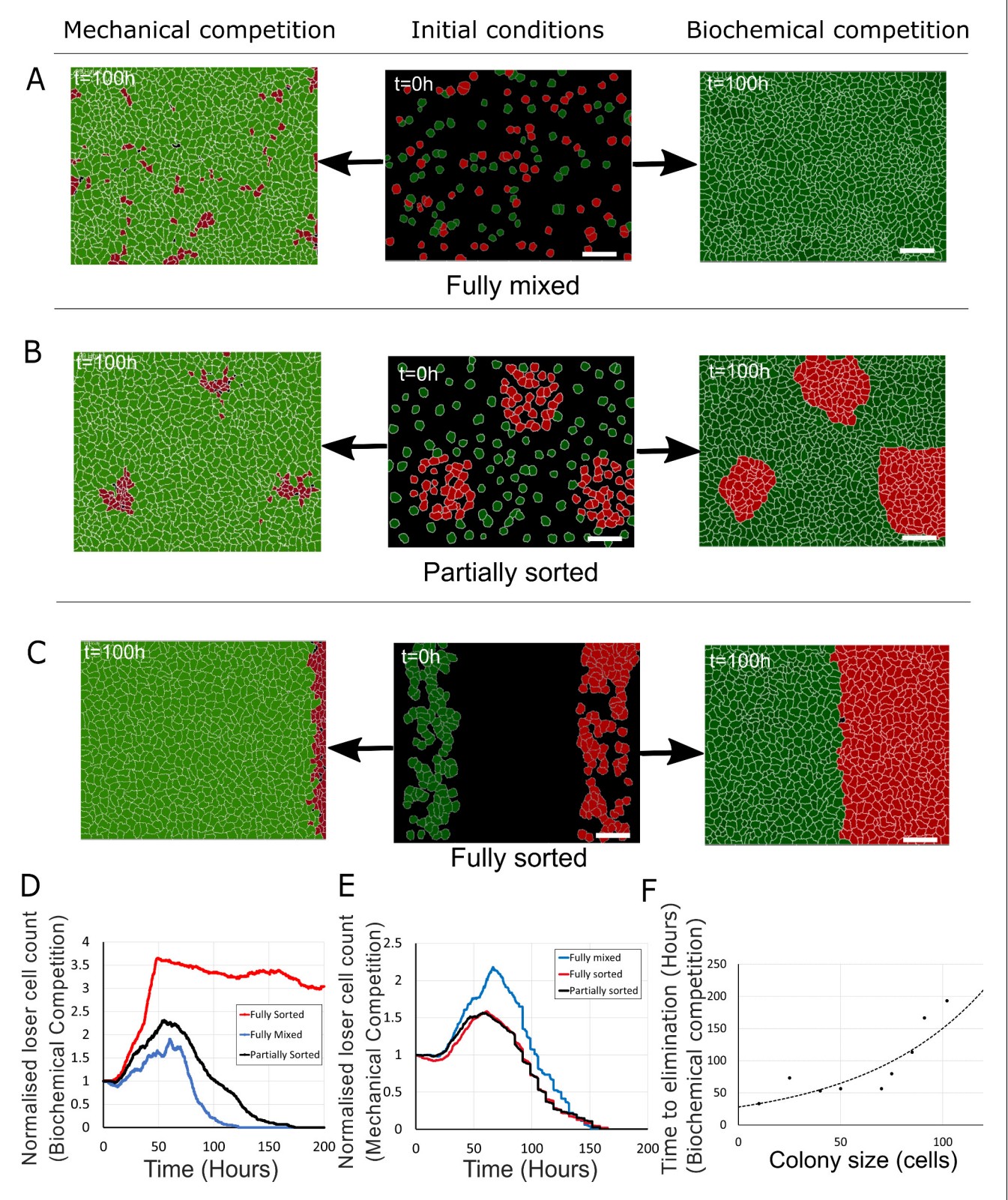

**Figure 6.** Tissue organisation governs the outcome of biochemical competition. (**A–C**) The middle panels show the initial configuration of a competition between 50% loser (red) and 50% winner (green) cells for various seeding arrangements (**A**: fully mixed; **B**: partially sorted; **C**: fully sorted). The right panels show a representative outcome for biochemical competition. The left panels show a representative outcome for mechanical competition. Cells are initially separated by free space (black). (**D**) Normalised loser cell count for the three configurations for biochemical competition

*Figure 6 continued on next page*

*Figure 6 continued*

(A–C, right-hand column). (E) Normalised loser cell count for the three configurations for mechanical competition (A–C, left-hand column). (D, E) Loser cell count for each configuration was averaged over three simulations. (F) Time to elimination as a function of the size of loser cell colonies for biochemical competition. Markers indicate the time determined in simulations for each colony size (N = 1 simulation for each colony size). The dashed line is a fit to an exponential function of the form $Ae^{-t/\tau}$, $r^2 = 0.73$ and $\tau \sim 60$ hr. Parameters used for the biochemical simulations in this figure are in *Supplementary file 3*.

The online version of this article includes the following figure supplement(s) for figure 6:

**Figure supplement 1.** Tissue organisation and heterotypic adhesion influence the outcome of competition in biochemical competition.

**Figure supplement 2.** Mechanical competition experiments in partially sorted starting conditions.

into a few colonies (*Figure 6B*, middle column, *Videos 7–9*), and fully sorted with loser cells and winner cells occupying opposite sides of the field of view (*Figure 6C*, middle column, *Video 10*). Strikingly, in mechanical competition, the normalised count of loser cells reached a maximum around 60 hr before continuously decreasing thereafter, consistent with our experimental observations in fully mixed (*Figure 4B*) and partially sorted conditions (*Figure 6—figure supplement 2A–C*, *Video 9*). By 160 hr, loser cells had been eliminated for all configurations (*Figure 6A–C*, left-hand column, E). In contrast, in biochemical competition, the outcome of competition appeared strongly dependent on initial seeding conditions with large differences in normalised count of losers after 100 hr (*Figure 6A–C*, right-hand column, D). Indeed, loser cell normalised count was close to 0 for fully mixed seeding but remained larger than 1 for partially and fully sorted seedings. By 200 hr, loser cell count had dropped to 0 in partially sorted seeding but only decreased gradually for fully sorted seeding (*Video 10*). This suggests that the kinetics of biochemical competition is sensitive to tissue organisation. To quantitatively compare tissue organisations, we computed the evolution of mixing entropy, a measure of local tissue organisation, in each competition (Materials and methods). We found that, when cells reached confluence, entropy of mixing was highest in the fully mixed seeding and lowest in the fully sorted seeding (*Figure 6—figure supplement 1A*). In the fully mixed and partially sorted seedings, mixing entropy decreased after overall confluence as the competition progressed, whereas for fully sorted seeding, mixing entropy stayed constant because the interface between winners and losers maintained its shape over time even though the number of loser cells gradually decreased (*Figure 6C*, *Figure 6—figure supplement 1A*, *Video 10*). Loser colony size and geometry may therefore determine the kinetics and outcome of biochemical competition. To gain further insight, we systematically varied the size of the loser colony and determined the time to elimination (*Figure 6F*). This revealed that the time to elimination monotonously increased with cell number in the colony but that no true steady-state coexistence was reached.

As we found that the kinetics of biochemical competition was controlled by tissue organisation and the intermixing of cells, we examined how the relative magnitude of heterotypic versus homotypic intercellular adhesion energy affects competition. In our simulations, we varied the heterotypic adhesion, $J_{heterotypic}$, while keeping $J_{homotypic}$ constant. In the Potts model, $J$ represents a surface energy which is the difference between the surface tension and adhesion. Therefore, when $J$ is high, intercellular adhesion is low. We found that the normalised loser cell count after 100 hr decreased with decreasing $J_{heterotypic}$ in fully mixed and partially sorted tissue organisation (*Figure 6—figure supplement 1C, D*). This is because for low $J_{heterotypic}$, cell intermixing is favoured and winner cells can invade colonies of loser cells, consistent with experimental observations (*Levayer et al., 2015*). However, in the fully sorted configuration, where the interface between the two cell types is minimal

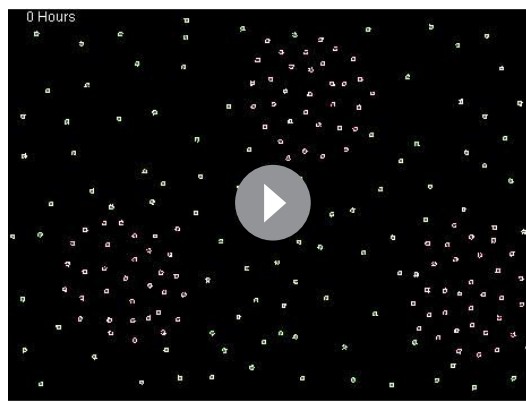

**Video 7.** Simulation of biochemical competition between 50% winner cell types and 50% loser cell types in an initial partially sorted configuration (*Figure 6B*).
https://elifesciences.org/articles/61011#video7

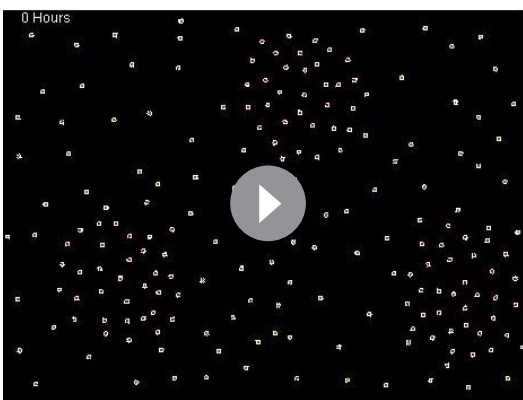

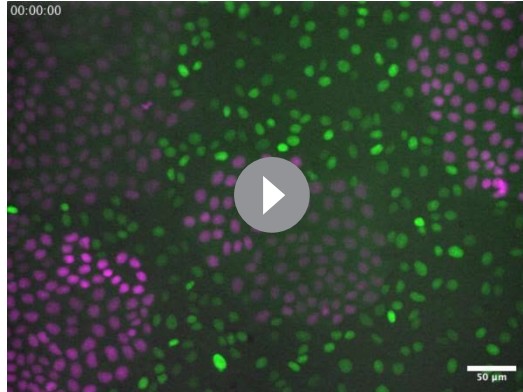

**Video 8.** Simulation of mechanical competition between 50% winner cell types and 50% loser cell types in an initial partially sorted configuration (*Figure 6B*).
https://elifesciences.org/articles/61011#video8

**Video 9.** Representative experiment of a competition between MDCK[WT] cells and MDCK[Scrib] cells in a partially sorted initial configuration. MDCK[WT] nuclei are marked with H2B-GFP (green), and MDCK[Scrib] nuclei are marked with H2B-mRFP (magenta). The movie represents 66 hr. Scale bar 50 μm.
https://elifesciences.org/articles/61011#video9

(*Figure 6C*), changes in $J_{heterotypic}$ only have a weak effect (*Figure 6—figure supplement 1B*). Thus, the kinetics of biochemical competition is sensitive to changes in parameters that lead to tissue reorganisation, such as the relative magnitude of homotypic and heterotypic adhesion energy.

In summary, our simple implementation of biochemical competition was sufficient to qualitatively reproduce current experimental observations, although the precise experimental curves relating probability of apoptosis to extent of heterotypic contact remain to be accurately determined experimentally.

## Discussion

In this study, we developed a multi-scale agent-based simulation to investigate of the interplay between physical cell interactions and probabilistic decision-making rules in deciding the outcome of cell competition. After parametrisation with experimental data, our model identified the physical and geometrical parameters that influence the outcome of mechanical and biochemical cell competition. Our analysis reveals that the kinetics of biochemical competition is governed by tissue organisation and parameters affecting it, whereas the outcome of mechanical competition is controlled by the difference in homeostatic density of the competing cell types together with energetic parameters.

Calibration of our model parameters separately for pure populations of winner and loser cells allowed us to quantitatively reproduce the experimentally measured kinetics of cell proliferation, mechanics of tissue homeostasis, the topology of tissue organisation, as well as the cumulative apoptosis and mitosis in each population. Winners and losers differed in their stiffness $\lambda$, their growth rates $G$, and their probability of apoptosis as a function of density. The latter is directly measured in our experiments (*Bove et al., 2017*) and is consistent with the increased sensitivity to crowding in loser cells due to interplay between stress pathways (*Wagstaff et al., 2016*; *Kucinski et al., 2017*).

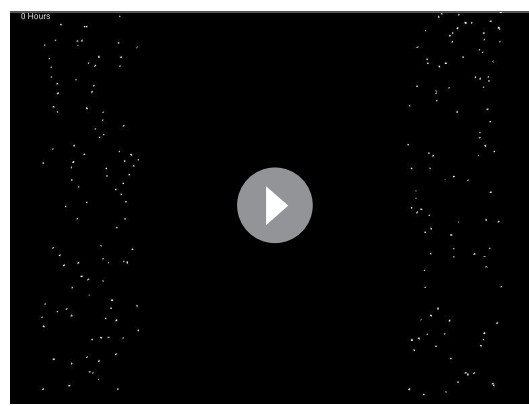

**Video 10.** Simulation of biochemical competition between 50% winner cell types and 50% loser cell types in an initial fully sorted configuration (*Figure 6C*).
https://elifesciences.org/articles/61011#video10

Our simulations showed that the growth rate $G$ controls the kinetics of competition but not its outcome. Overall, only two parameters governed the outcome of mechanical competition: the stiffness $\lambda$ and the sensitivity to contact inhibition quantified by the parameter $k$ (*Supplementary file 4*).

The contact inhibition parameter $k$ regulates the rate of cell growth and, consequently, the cell cycle duration (Materials and methods). In winner cells, contact inhibition of growth controls homeostatic density, which is lower for high contact inhibition, $k$. However, $k$ does not control homeostatic density in loser cells due to their increased probability of apoptosis under even moderate crowding conditions. As a result, when $k$ is increased, the difference in homeostatic density between the winners and losers, $\Delta HD$, decreases. This slows the kinetics of mechanical competition, eventually abolishing it. Thus, our simulations predict that $\Delta HD$, which is related to the difference in homeostatic density (*Basan et al., 2009*), is a good predictor for the outcome of mechanical competition and therefore that perturbing the molecular mechanisms that participate in setting cellular homeostatic density should alter the outcome of competition (*Eisenhoffer et al., 2012*; *Gudipaty et al., 2017*). A further implication is that mechanical competition may actually represent a cell-autonomous phenomenon where each cell type independently seeks to reach its homeostatic density in a dynamically changing environment.

Our model parametrisation based on experimental data suggests that loser cells are typically more compressible (or softer) than winners. As a consequence, in competition assays, loser cells tend to decrease their apical areas more than winners after confluence, as observed in experiments (*Wagstaff et al., 2016*). This higher local density, together with losers' higher sensitivity to crowding, results in preferential elimination of loser cells. Conversely, when losers are stiffer than winners, their local density does not increase dramatically and they survive. Thus, the relative stiffness parameter $\Lambda$ emerges as a key control parameter for mechanical competition. Loser cells tend to be eliminated if $\Lambda<1$, whereas they survive for $\Lambda>1$. Although by convention $\lambda$ is referred to as an area expansion modulus, cells are 3D objects and their volume is tightly regulated even when subjected to mechanical deformations (*Harris and Charras, 2011*; *Harris et al., 2012*). Thus, the decrease in apical area of loser cells in competition implies a concomitant increase in their height, consistent with experimental observations (*Wagstaff et al., 2016*). Therefore, $\lambda$ could be interpreted as a height elastic modulus that may emerge from the ratio of apical to lateral contractility that governs the height of epithelial cells in 3D vertex models (*Latorre et al., 2018*). Overall, both contact inhibition of proliferation and planar cell compressibility altered the outcome of mechanical competition by changing the local density attained by the loser cells. Thus, mechanical competition appears to be primarily regulated by parameters controlling the compressional mechanical energy stored in the system.

Our previous work revealed that division of MDCK[WT] cells is favoured in neighbourhoods with many MDCK[Scrib] cells, potentially indicating an inductive behaviour (*Bove et al., 2017*). However, our simulations of mechanical competition based solely on differences in sensitivity to crowding also revealed that winner cells are more likely to divide when in contact with loser cells. Therefore, the upregulation of winner cell division in loser neighbourhoods represents an emergent property of our simulation, likely arising from the combination of a higher growth rate and a lower sensitivity to crowding in winner cells. However, the magnitude of this effect was smaller than in experiments, perhaps pointing to a role for signalling mechanisms accelerating the cell cycle in response to free space that was observed in experiments (*Gudipaty et al., 2017*; *Streichan et al., 2014*) but not implemented in our simulations. Overall, our simulations suggest that the interaction between MDCK[WT] and MDCK[Scrib] cells in our experiments can be entirely explained by mechanical competition alone despite suggestions that signalling mechanisms such as active corralling of losers cells may play a role (*Wagstaff et al., 2016*).

Biochemical competition depends on the extent of heterotypic contact between losers and winners. As a result, the kinetics of biochemical competition strongly depends on tissue organisation but is not affected by changes in cell compressibility or contact inhibition (*Supplementary file 5*). Instead, two parameters controlled biochemical competition: the heterotypic contact energy and the initial organisation of the tissue. Indeed, tissue organisations with greater mixing between the cell types resulted in more rapid elimination of the loser cells (*Figure 6*). This arises as a natural consequence of the probability of apoptosis of loser cells depending on the extent of heterotypic contact. In other words, competition depends on the extent of cell intermixing. Consistent with this, when the heterotypic contact energy was lower

than the homotypic contact energy, this led to more mixing between cell types and more cell competition. Interestingly, experimental evidence has revealed that perturbations that promote cohesion of losers protect against elimination, while those that promote intercalation of winners and losers lead to greater loser elimination (*Levayer et al., 2015*). Our study only examined varying $J_{heterotypic}$ in conditions where $J_{homotypic}$ was the same in both cell types. However, other conditions can also occur. In particular, previous work has shown that clone shape and mixing is set by the ratio of tension within and outside of the clone (*Bosveld et al., 2016*). This paints a picture where loser cells can mix extensively with the winner cells as long as loser cell cohesion is sufficiently strong to prevent intrusion of winner cells into loser cell colonies (i.e., $J_{homotypic, winner} > J_{homotypic, loser}$ and $J_{heterotypic}$). While cells did not possess high motility in our simulations, we would expect this to affect the outcome of biochemical competition as motility would increase cell intermixing. The initial organisation of the tissue strongly influenced the kinetics of elimination, and our simulations revealed a steady increase in the time to elimination with increasing number of cells in the loser colony but no steady coexistence. However, in practice, above a certain colony size, the time to elimination will exceed the lifespan of the organism, signifying that coexistence takes place *de facto*. In our simulations, winner and loser cells had an identical growth and therefore a higher growth rate in the loser cells could in theory be sufficient to ensure a regime with coexistence of the two cell types. However, theoretical considerations show that such a regime is unstable and will lead to elimination of one or the other cell types in response to small perturbations in growth rate or colony size. Intriguingly, experiments in vivo have revealed a regime of coexistence (*Levayer et al., 2015*). Many hypotheses could explain this discrepancy, for example, a different probability of apoptosis function than the one implemented here or the existence of additional signalling mechanisms not considered in our simple simulation. An in-depth study of the conditions for coexistence will form an interesting direction for future research as this occurs in many biological tissues and will be greatly enhanced by experimental determination of the function relating probability of apoptosis to the fraction of the loser cell perimeter in contact with winner cells.

In summary, our study revealed that mechanical competition is governed by factors that reduce the stored mechanical energy in the system, while biochemical competition is favoured by factors that increase cell intermixing and tissue reorganisation. Conversely, mechanical competition was not affected by tissue organisation, whereas biochemical competition was not sensitive to parameters that changed the stored mechanical energy of the tissue.

## Materials and methods

### Cellular Potts model

The CPM is implemented in Compucell3D (*Swat et al., 2012*). We chose a 2D lattice-based model, where cells are composed of a collection of lattice sites (pixels). Cells interact at their interfaces through predefined adhesion energies, and several different cell types can be implemented. Each cell is then given attributes characterising their mechanical and adhesive properties. For example, each cell is assigned a cell type $\tau$, which in turn has some value of surface contact energy $J_{cell-cell}$ with other cell types and adhesion energy $J_{cell-substrate}$ with the substrate. Cells are also assigned a target area $A_T(t)$ (that represents the area a cell would occupy at time $t$ if it were isolated) and an area expansion modulus $\lambda$ (that represents the energetic cost of increasing cell area and originates from the mechanical properties of the cytoskeleton). In the computational cell decision-making in our simulation, $A_T$ and $\lambda$ play important roles in the implementation of growth and division dynamics. In addition, we also incorporate active cell motility. The free energy of the system is given by the Hamiltonian $H$:

$$H = \sum_{<i,j>} J\left(\tau\left(\sigma_{ij}^k\right), \tau\left(\sigma_{i'j}\right)\right)\left(1 - \delta\left(\sigma_{ij}^k, \sigma_{i'j}\right)\right) + \lambda \sum_{\sigma}\left(A(\sigma^k) - A_T(\sigma^k)\right)^2 \Theta(\tau) + \lambda_m \sum_{\sigma} \hat{m}(\sigma, t) \cdot \hat{s},$$

where the first term describes the interaction of lattice sites due to the adhesion energy between the cell types or between cells and the substrate. The coefficient $J$ is the surface energy between cell type $\tau$ of the target lattice site $\sigma^k$ and the cell type $\tau\left(\sigma_{i'j}\right)$ of its nearest-neighbour lattice

points. By convention, for free space $\tau = 0$. From a biological perspective, $J$ is the difference between the surface tension and intercellular adhesion. Therefore, higher $J$ implies lower intercellular adhesion. The multiplicative term $\left(1 - \delta\left(\sigma_{ij}^k, \sigma_{i'j'}\right)\right)$ prevents cells from interacting energetically with themselves, where

$$\delta\left(\sigma_{ij}^k, \sigma_{i'j'}\right) = \begin{cases} 1, & \sigma_{ij}^k = \sigma_{i'j'}, \\ 0, & \sigma_{ij}^k \neq \sigma_{i'j'}. \end{cases}$$

The second term in the Hamiltonian describes an additional energy cost due to deviation of the actual area $A(\sigma)$ of a cell from its target area $A_T(\sigma)$, specific to each cell at time $t$. In the second computational layer of the simulation, $A_T$ is varied at each time point to reflect cell growth. The coefficient $\lambda$ represents the area expansion modulus in 2D, which is related to planar cell stiffness or the ratio between apical and lateral contractility that control cell height. We introduce the term $\Theta(\tau)$ to treat the free space pixels differently from pixels belonging to cells. In contrast to cells, the free space does not have a target area, and hence no associated mechanical energy.

$$\Theta_\tau = \begin{cases} 0, & \tau\left(\sigma_{ij}^k\right) = 0 \text{(freespace)}, \\ 1, & \text{otherwise}. \end{cases}$$

The final term in the Hamiltonian assigns active motility to the cells along a random unit vector $m$ (*Li and Lowengrub, 2014*). Here, $s$ is the spin flip direction between the lattice site in question and one of its neighbouring lattice sites.

## Model parametrisation

To describe epithelial cell dynamics using the Potts model, we parametrised it using our experimental data (*Bove et al., 2017*). For simplicity, we chose the same length scale for pixels in our simulation as in our experimental images of competition experiments. The lattice size and cell sizes are chosen to match the experimental data. The lattice is chosen to be $1200 \times 1600$ pixels, where each pixel is $0.33 \times 0.33 \ \mu m^2$.

The target area and stiffness of each cell type were determined based on the average cell areas measured from cells isolated from one another in brightfield images (*Figure 2—figure supplement 2*).

In the simulation, one Monte Carlo timestep (MCS) is defined by each lattice point being given the possibility of changing identity. A conversion between experimental time and computational time was derived empirically by comparing the mean squared displacements of isolated cells in experiments to those in the simulations. We found that 10 MCS represented one frame of a time-lapse movie in our experiments (4 min).

## Cell growth and division

The agent-based part of the model requires the introduction of cellular behaviour in the form of probabilistic rules for cell growth, division, extrusion, and apoptosis. In our simulations, cells grow linearly by increasing their target areas $A_T(t)$ at a rate G, which was chosen to replicate the average cell doubling time measured in experiments pre-confluence (*Bove et al., 2017*). In line with recent experimental work (*Cadart et al., 2018*), we assume that MDCK cells follow an 'adder' mechanism for cell size control, such that cells divide along their major axis once a threshold volume $\Delta A_{tot}$ has been added since birth (*Figure 1—figure supplement 1*, *Figure 1—figure supplement 3*). In our simulations, the added cell volume at each time point was a random value distributed around the mean experimental value G, so as to capture cell-to-cell variability. When the simulation was initialised, cell areas had a homogenous distribution to mimic a uniform probability for cells of being in any given stage of their cell cycle at the start of experiments.

## Contact inhibition of proliferation

In our simulations, cells possess a target area, $A_T(t)$, which they would occupy at that time if they had no neighbours, and an actual area, $A(t)$, which they currently occupy. As $A_T$ increases at each timestep due to cell growth, the difference between their target and actual area $A$ increases. If this

difference becomes too large, the second term of the Hamiltonian dominates, leading to energetically unfavourable swaps and a collapse of the network. To mimic reduced protein synthesis reported due to contact inhibition of proliferation (*Azar et al., 2010*), we assume that the effective growth rate depends on the difference between $A$ and $A_T$:

$$\frac{dA_T}{dt} = Ge^{-k(A-A_T)^2},$$

where $G$ is the growth rate for cells with no neighbours, $A_T$ the target cell area, $A$ the actual area, and $k$ quantifies the sensitivity to contact inhibition. $k$ parametrises how much deviation can be tolerated between the target area and current cell area before growth stalls. Note that this condition is applied iteratively at every frame for each cell, such that, when free space becomes available, growth can immediately resume nearby.

## Apoptosis due to competition

In crowded conditions such as those present in mechanical competition, the probability of apoptosis increases with local cell density (*Wagstaff et al., 2016*; *Bove et al., 2017*; *Eisenhoffer et al., 2012*). To implement this, each cell was assigned a probability of apoptosis $p_{apo}$ at each timestep that depended on its local cell density $\rho$. In our simulation, ρ was defined as the sum of inverse of areas of the cell of interest $\sigma^k$ and its first neighbours $\sigma^i$: $\rho(\sigma^k) = \frac{1}{A(\sigma^k)} + \sum_{i=1}^{n} \frac{1}{A(\sigma^i)}$ (*Figure 1—figure supplement 4B*). Based on our experimental data, we decided to describe the relationship between $p_{apo}$ and $\rho$ as: $p_{apo}(\rho) = \frac{p_{apo,max}}{\left(1 + e^{-\alpha\left(\rho - \rho_{1/2}\right)}\right)}$ (*Figure 1—figure supplement 4A*). $p_{apo,max}$ was fixed to the same value for both populations, and $\rho_{1/2}$ was determined for each population separately based on experimental data (*Bove et al., 2017*).

In biochemical competition, apoptosis occurs when loser cells are in direct contact with winner cells. Recent work has shown that in *Drosophila* the probability of apoptosis of loser cells depends on the percentage of the perimeter in contact with the winner cells (*Levayer et al., 2015*). Following this, we chose to implement the probability of apoptosis as a sigmoid function (Hill function) following the relationship $p_{apo}(p) = p_{apo,max} p^n / (S^n + p^n)$, where $p$ is the percentage of perimeter in heterotypic contact, $n$ is the Hill coefficient, $p_{apo,max}$ is the maximum probability, and $S$ is the steepness. We chose a maximum probability $p_{apo,max}$ of death per frame similar to that encountered in mechanical competition and a Hill coefficient $n = 3$ (*Figure 1—figure supplement 5A*). This is justified by the fact that mechanical and biochemical competition take place over comparable durations in MDCK cells ~2–4 days (*Hogan et al., 2009*; *Kajita et al., 2010*). For both mechanical and biochemical competition, the execution of apoptosis was implemented by setting the target area $A_T$ of the cell to 0 and area expansion modulus λ to 2. This allows for a quick but not instantaneous decrease of the cell area until the cell is completely removed.

## Live extrusion of cells

Under conditions where the local cell density increases rapidly, live cells can be extruded from monolayers, likely because they have insufficient adhesion with the substrate to remain in the tissue (*Eisenhoffer et al., 2012*; *Kocgozlu et al., 2016*; *Figure 1—figure supplement 1D*). We assumed that cells underwent live extrusion when their area $A_i$ was $A_i \leq \langle A(t) \rangle / 2$ with $<A(t)>$ the average area of all cells in the simulation. Once this occurs, the cell is eliminated immediately from the tissue. Unlike apoptosis, cell elimination via extrusion is implemented as an instantaneous removal of the qualifying cell from the lattice to reflect the faster rate of live extrusions compared to programmed cell death.

## Acquisition and analysis of experimental data

All simulated data for mechanical competition were compared quantitatively to experiments acquired in *Bove et al., 2017* or performed specifically for this publication. Methods for cell culture, image acquisition, segmentation, and analysis are described in detail in *Bove et al., 2017*. Briefly, MDCK wild-type cells (MDCK$^{WT}$) were winners in these competitions and their nuclei were labelled with H2B-GFP, while MDCK scribble knock-down cells (MDCK$^{Scrib}$, described in *Norman et al., 2012*) were the losers and labelled with H2B-RFP (*Bove et al., 2017*). MDCK$^{Scrib}$ cells conditionally

expressed an shRNA targeting scribble that could be induced by addition of doxycycline to the culture medium. All cell lines were regularly tested for mycoplasma infection and were found to be negative (MycoAlert Plus Detection Kit, Lonza, LT07-710).

MDCK$^{WT}$ cells were grown in DMEM (Thermo Fisher) supplemented with 10% fetal bovine serum (Sigma-Aldrich), HEPES buffer (Sigma-Aldrich), and 1% penicillin/streptomycin in a humidified incubator at 37°C with 5% $CO_2$. MDCK$^{Scrib}$ cells were cultured as MDCK$^{WT}$, except that we included tetracycline-free bovine serum (Clontech, 631106) to supplement the culture medium. To induce expression of scribble shRNA, doxycycline (Sigma-Aldrich, D9891) was added to the medium at a final concentration of 1 μg/ml.

For competition assays, winner and loser cells were seeded in the chosen proportion to reach an overall density of 0.07 cells per 100 μm$^2$ and left to adhere for 2 hr. Cells were then imaged every 4 min for 4 days using a custom-built incubator microscope and the appropriate wavelengths (*Bove et al., 2017*). Movies were then automatically analysed to track the position, state, and lineage of the cells using deep-learning-based image classification and single-cell tracking as detailed in *Bove et al., 2017*. Single-cell tracking was performed using bTrack (github.com/quantumjot/BayesianTracker; *Lowe, 2021*). Cell neighbours are determined using a Voronoi tessellation dual to the Delaunay triangulation of nuclei.

In most experiments, knock-down of scribble was induced by addition of doxycycline 48 hr before the beginning of the experiment to ensure complete depletion. However, for experiments examining cell competition in partially sorted conditions, we first seeded the same number of MDCK$^{Scrib}$ cells as in 90:10 competitions and cultured them for 48 hr without doxycycline until they formed well-defined colonies. Then, we added the number of MDCK$^{WT}$ cells that would be expected after 48 hr competition, left them to adhere for 4 hr, and added doxycycline. We then started imaging the following day, signifying that scribble depletion was incomplete at the start of the experiment.

### Cell fate analysis
For the analysis of experiments and simulations, fate information for each cell is dynamically recorded to a file and analysed using a custom software written in Matlab (*Bove et al., 2017*).

### Entropy of cell mixing
The entropy of mixing was calculated as the Shannon entropy of a two-state system, where the states considered are the cell types (winner/loser). The entropy was then calculated as $s = -P_1 \ln P_1 - P_2 \ln P_2$ for each cell at each frame, where $P_1 = \frac{\# \ winner \ neighbors}{total \ \# \ neighbors}$ and $P_2 = \frac{\# \ loser \ neighbors}{total \ \# \ neighbors}$. The entropy of the whole system was then calculated as $S = <s>/\sum cells$.

### Probability estimation
To calculate the probability of apoptosis and division for each cell type, cells were binned appropriately (by density or by time). Then, we determined the number of events $n$ (apoptosis or division) and the total number of cells of each type $N$ (the observations) in each bin. The probability $p$ was then computed as $p = \frac{n}{N}$. Because we are examining rare events, we calculated the coefficient of variation $cv$ that measures the relative precision of our estimator of probability as $cv = \sqrt{\frac{(1-p)}{pn}}$.

### Data and model availability
Our model has been deposited in Github (https://github.com/DGradeci/cell_competition_paper_models; copy archived at swh:1:rev:55f8b189c6f5d998cc5b2819f672ad80b547c956; *Gradeci, 2021*). The data used for model calibration will be deposited in doi: 10.5522/04/12287465.

## Acknowledgements
This work was supported by Engineering and Physical Sciences Research Council (EPSRC) PhD studentships to DG and AB. SB acknowledges support from the Royal Society (URF/R1/180187). GV was supported by BBSRC grant BB/S009329/1 to AL and GC. The authors also wish to acknowledge the reviewers whose insightful comments greatly improved the study.

## Additional information

### Funding

| Funder | Grant reference number | Author |
|---|---|---|
| Biotechnology and Biological Sciences Research Council | BB/S009329/1 | Alan R Lowe<br>Guillaume Charras |
| Royal Society | URF | Shiladitya Banerjee |
| Engineering and Physical Sciences Research Council | PhD studentships | Daniel Gradeci<br>Anna Bove |

The funders had no role in study design, data collection and interpretation, or the decision to submit the work for publication.

### Author contributions

Daniel Gradeci, Data curation, Software, Formal analysis, Methodology, Writing - original draft, Writing - review and editing; Anna Bove, Data curation, Formal analysis, Investigation, Methodology; Giulia Vallardi, Data curation, Investigation, Writing - review and editing; Alan R Lowe, Conceptualization, Software, Supervision, Funding acquisition, Writing - original draft, Project administration, Writing - review and editing; Shiladitya Banerjee, Conceptualization, Software, Supervision, Writing - original draft, Writing - review and editing; Guillaume Charras, Conceptualization, Supervision, Funding acquisition, Writing - original draft, Writing - review and editing

### Author ORCIDs

Alan R Lowe ⓘ https://orcid.org/0000-0002-0558-3597
Shiladitya Banerjee ⓘ https://orcid.org/0000-0001-8000-2556
Guillaume Charras ⓘ https://orcid.org/0000-0002-7902-0279

### Decision letter and Author response

Decision letter https://doi.org/10.7554/eLife.61011.sa1
Author response https://doi.org/10.7554/eLife.61011.sa2

## Additional files

### Supplementary files

• Supplementary file 1. Table of default parameters for mechanical competition in *Figures 2* and *4*. Parameters for the Potts model are shaded in grey. Initial conditions and computational implementation parameters are in white. Cell automaton parameters are in pink. Probability of apoptosis is implemented by a function of the form $p_{apo}(\rho) = \frac{p_{apomax}}{\left(1+e^{-\alpha\left(\rho-\rho_{1/2}\right)}\right)}$. A.u.: arbitrary units.

• Supplementary file 2. Table of parameters for investigating the control of mechanical competition in *Figure 5* and *Figure 5—figure supplement 1*. Parameters for the Potts model are shaded in grey. Initial conditions and computational implementation parameters are in white. Cell automaton parameters are in pink. Parameters that were varied are indicated, and the range is given in brackets. Probability of apoptosis is implemented by a function of the form $p_{apo}(\rho) = \frac{p_{apomax}}{\left(1+e^{-\alpha\left(\rho-\rho_{1/2}\right)}\right)}$. A.u.: arbitrary units.

• Supplementary file 3. Table of parameters for investigating the control of biomechanical competition in *Figure 6* and *Figure 1—figure supplement 5*. Parameters for the Potts model are shaded in grey. Initial conditions and computational implementation parameters are in white. Cell automaton parameters are in pink. Parameters that were varied are indicated, and the range is given in brackets. Probability of apoptosis is implemented by a Hill function of the form $p_{apo}(p) = p_{apomax}\, p^n/(S^n + p^n)$ with $p$ the fraction of perimeter occupied by heterotypic adhesions. A. u.: arbitrary units.

- Supplementary file 4. Table of parameters investigated in simulations of mechanical competitions: fixed parameters. See *Supplementary files 1* and *2* for exact values.

- Supplementary file 5. Table of parameters investigated in simulations of biochemical competitions: fixed parameters. See *Supplementary file 3* for exact values. Note that all density-related parameters are fixed because apoptosis does not depend on density.

- Transparent reporting form

## Data availability

The data used for model calibration will be deposited on UCL's research data repository (https://rdr.ucl.ac.uk/). The following doi has been reserved for publication: 10.5522/04/12287465. Our model has been deposited in Github (https://github.com/DGradeci/cell_competition_paper_models; copy archived at https://archive.softwareheritage.org/swh:1:rev:55f8b189c6f5d998cc5b2819f672ad80b547c956).

The following dataset was generated:

| Author(s) | Year | Dataset title | Dataset URL | Database and Identifier |
|---|---|---|---|---|
| Gradeci D, Vallardi G, Banerjee S, Charras G | Bove A, Lowe A, | 2021 | Cell-scale biophysical determinants of cell competition in epithelia | https://doi.org/10.5522/04/12287465 | University College London, 10.5522/04/12287465 |

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
