## [Decision Letter]

**Acceptance summary:**

The authors study cell competition using simulations and experiments. The simulations quantitatively reproduce experiments and show that mechanical competition is determined by differences between the homeostatic densities of winners and losers, whereas tissue organization is key for biochemical competition. By linking cell-scale mechanisms to tissue-scale organization, the study provides fundamental insights into mechanisms underlying cell competition and is of broad interest for developmental biologists

**Decision letter after peer review:**

Thank you for submitting your article "Cell-scale biophysical determinants of cell competition in epithelia" for consideration by *eLife*. Your article has been reviewed by 2 peer reviewers, one of whom is a member of our Board of Reviewing Editors, and the evaluation has been overseen by Naama Barkai as the Senior Editor. The following individual involved in review of your submission has agreed to reveal their identity: Romain Levayer (Reviewer #2).

The reviewers have discussed the reviews with one another and the Reviewing Editor has drafted this decision to help you prepare a revised submission.

Summary:

In this manuscript, Gradeci et al. use a theoretical approach together with experiments to investigate biophysical determinants of cell competition in epithelia. Employing a cellular Potts model, which is fitted to rich experimental data, they determine values of parameters quantifying mechanical and biochemical processes involved in cell competition. The major finding of this work is that the difference between the homeostatic density of the winner and loser cell populations is a key mechanical parameter affecting competition. However, it is poorly affected by growth rate or tissue architecture. While, for example, Basan et al. 2009 already suggested that differences in growth rate will poorly affected the outcome of competition based on differences in homeostatic pressure, this is the first time that it is clearly demonstrated that they are sufficient to reproduce quantitatively experimental data. The authors also provide evidence that biochemical competition strongly depends on tissue organization. It is one of the most realistic modeling of cell competition which has been performed so far.

Essential revisions:

1. The manuscript seems to be poorly written. There is not a nice logical flow and the figure panels are referred to in an almost random manner, which makes it very hard to understand this work. Also, Figure 1A,B,E and Figure 2 are not very helpful. Several assumptions are not clearly explained. Please, improve the text.

2. The Potts model should be related more closely to cellular properties. What do the parameters represent? Can you give us information about how sensitive your results are to variations in the parameter values?

Previously, the authors have shown that WT cell proliferation increases in the vicinity of Scribble mutant cells (Bove et al. MBOC 2017). Using a continuous model, they also proposed that this boost of proliferation is required to recapitulate the dynamics of the two populations. It is somehow surprising that the authors did not describe or implement this process in their cellular Potts model (which now can includes information about cellular neighbourhood). Indeed, while the difference in homeostatic density is sufficient to recapitulate the cell population dynamics, it does not prove that this is the actual mechanism at play, and does not exclude alternative mechanisms. Could the authors test whether this alternative mechanism could – or could not – be sufficient to recapitulate Scribble competition dynamics? More generally, the point raised by the authors could be much stronger if they could compare the accuracy of their current model (purely based on differences in homeostatic density) with alternative models (e.g.: non cell autonomous process) to reproduce the Scribble competition scenario.

Also, alternatively, could this boost of proliferation be an emerging feature of differences in stiffness and local increase of Scribble elimination (hence increasing locally WT cell area and their proliferation )? Do the authors observe such local boost of proliferation in their model without implementing additional rules ?

3. So far, the main outputs of the model compared with the experiment are the evolution of cell density and number of cells. However, the authors do have experimental data about the rate of death and rate of division (Bove et al. 2017, Figure 2G). Actually the cumulative rate of apoptosis obtained in the simulation (Figure S4D of this study) seems to be different from the experimental curves (cell death of Scribble cells raised later in experiments, and the difference with WT cells is not as strong). Could the author comment on that or try to find an explanation ?

4. Most of the time, the authors mention the disappearance of the loser cells in the text, however most of the simulations finish before full disappearance of the loser cells (e.g. Figure 6C, k=0.1 and 1). Is this a matter of time (longer simulation would lead to full disappearance) or is there a steady state with loser cells maintained at low number ?

5. As stated and shown by the authors, the size of the cluster of loser cells strongly influence the outcome of biochemical competition. It is striking that for the fully sorted conditions, the losers survive irrespective of J-heterotipic. Intuitively this might be related to a perimeter other area ratio which reaches a critical value where apoptosis rate / cell splitting rate (both scaling with perimeter) are always lower than proliferation (scaling with area). Do you also observe such critical cluster size appearing in the partially sorted conditions ? This may be reflected by the final distribution of loser cell-clusters in the partially sorted condition (all clusters being larger than this critical value).

---

## [Author Response]

Essential revisions:1. The manuscript seems to be poorly written. There is not a nice logical flow and the figure panels are referred to in an almost random manner, which makes it very hard to understand this work. Also, Figure 1A,B,E and Figure 2 are not very helpful. Several assumptions are not clearly explained. Please, improve the text.

After carefully re-reading our manuscript, we agree that the logical flow and figure numbering needed improvement. We have now revised the manuscript paying close attention to the logical flow, clarity and explanation of assumptions, as well as figure referencing. We now hope that the revised article is easier to follow and understand.

In particular, following the reviewers’ suggestion, we have replaced Figures 1-2 by a new figure. After careful consideration, we have decided to keep this figure in the main text as it gives non-specialist readers a graphic overview of the principle of the simulations and the flow of information.

2. The Potts model should be related more closely to cellular properties. What do the parameters represent? Can you give us information about how sensitive your results are to variations in the parameter values?

We apologise that this was not clear in the original version of our manuscript.

In our implementation, each cell is described by a cell type τ, a preferred area AT that it would occupy in isolation, and an actual area A that it occupies. Both A and AT depend on time.

The Potts model relies on four mechanical parameters to describe interactions between a cell and its environment: J_homotypic_, J_heterotypic_, J_cell-substrate_, and λ. J_homotypic_ describes the adhesive interaction of each cell with cells of the same type, J_heterotypic_ describes adhesive interaction between cells of different type, and J_cell-substrate_ describes the adhesive interaction with the extracellular matrix. All of these parameters define adhesion energies and therefore a lower energy value implies a stronger adhesion. The parameter λ represents the stiffness of the cell controlled by the actomyosin cytoskeleton. No experimental measurements currently exist for these parameters and therefore, they need to be calibrated empirically by comparing the output of simulations to experimental data.

In addition, in our model, a cell automaton implements cell growth, cell division, and cell death. This cell automaton necessitates a number of other parameters to be calibrated. Cells are born with a size AT(0) and, at each timestep of their life, their target size is increased at a rate dA_T_/dt that depends indirectly on local crowding of the epithelium to reflect contact inhibition of proliferation. In our simulation, the rate of increase in target area is dATdt=Ge−k(A(t)−AT(t))2 with G the maximum growth rate observed in sub-confluent conditions and k a heuristic parameter implementing contact inhibition. Based on data demonstrating that MDCK cells grow following an adder model (Cadart et al., Nat Comms, 2018), cells divide when they have added a total area ΔAtot to their target area during the cell cycle. A_T_(0), G, and ΔAtot can all be measured directly from experimental data. The contact inhibition parameter *k* is calibrated based on the cell density at homeostasis. Conversely, at each timestep, cells have a probability of apoptosis p_apo_ that depends on the local density ρ for mechanical competition. A relationship between p_apo_ and ρ was determined experimentally in competition data from a previous publication (Bove et al., MBoC, 2017). In biochemical competition, no such relationship has been determined experimentally and we chose a relationship on the basis of qualitative work in the field.

In our simulations, we chose to fix some of the parameters because they are directly measurable. For example, we fixed A_T_(0), ΔAtot, and p_apo_ as a function of ρ in mechanical competition. Other parameters were calibrated by comparison of simulations to experiments. To examine how sensitive our results are to each parameter, we varied all the undetermined parameters in our simulations to investigate how they affect the kinetics and outcome of competition. In mechanical competition, we varied k, λ, G, J_heterotypic_ and the initial cell configuration. In biochemical competition, we varied J_heterotypic_, the initial cell configuration, and the sensitivity of apoptosis to heterotypic contacts. The nature of each parameter and whether it was varied or not is now summarised in two supplementary tables: **Supplementary Files 4 and 5**.

These clarifications and a summary of the influence of each parameter on the outcome of cell competition are now discussed in the main text and in the Discussion section.

Previously, the authors have shown that WT cell proliferation increases in the vicinity of Scribble mutant cells (Bove et al. MBOC 2017). Using a continuous model, they also proposed that this boost of proliferation is required to recapitulate the dynamics of the two populations. It is somehow surprising that the authors did not describe or implement this process in their cellular Potts model (which now can includes information about cellular neighbourhood). Indeed, while the difference in homeostatic density is sufficient to recapitulate the cell population dynamics, it does not prove that this is the actual mechanism at play, and does not exclude alternative mechanisms. Could the authors test whether this alternative mechanism could – or could not – be sufficient to recapitulate Scribble competition dynamics? More generally, the point raised by the authors could be much stronger if they could compare the accuracy of their current model (purely based on differences in homeostatic density) with alternative models (e.g.: non cell autonomous process) to reproduce the Scribble competition scenario. Also, alternatively, could this boost of proliferation be an emerging feature of differences in stiffness and local increase of Scribble elimination (hence increasing locally WT cell area and their proliferation )? Do the authors observe such local boost of proliferation in their model without implementing additional rules ?

This is a very interesting point and we would like to thank the reviewer for bringing this up. New analysis of our simulations revealed a local boost of proliferation of winner cells in the neighbourhood of loser cells, as an emergent property of our simulation without requiring implementation of additional rules.

To address the reviewers’ question, we compared the probability of division for winner cells in contact with loser cells and those in contact only with winner cells, as suggested by the reviewers. We found that winner cells in contact with loser cells had a 25% higher probability of division than when they were not in contact with winners (with a coefficient of variation of less than 5%). Therefore, it appears that the increased probability of division for winner cells surrounded by loser cells is an emergent property of our simulation. Conceptually, this phenomenon results from the combination of a higher probability of apoptosis in loser cells together with a lower growth rate. Thus, free space is more likely to arise when a winner cell is in contact with a loser cell and the winner cell is also more likely to take advantage of this opportunity because of its larger growth rate. We do however note that the magnitude of the increase is smaller than what we observed experiments. This is likely because signalling mechanisms not implemented in our simulations amplify the effect. In our view, without further experimentation, implementing specific mechanisms to exactly match the increase in growth rates would not yield any further insight. This data had now been added to the main text and in **Figure 4G.**

3. So far, the main outputs of the model compared with the experiment are the evolution of cell density and number of cells. However, the authors do have experimental data about the rate of death and rate of division (Bove et al. 2017, Figure 2G). Actually the cumulative rate of apoptosis obtained in the simulation (Figure S4D of this study) seems to be different from the experimental curves (cell death of Scribble cells raised later in experiments, and the difference with WT cells is not as strong). Could the author comment on that or try to find an explanation ?

We would like to thank the reviewers for this suggestion. Upon rereading our manuscript, we agree that additional metrics should be used to compare simulations with experiments. We have now added several further points of comparison between our simulations and our experiments. First, we computed the probability of apoptosis as a function of density for each cell type and verified that this closely matched our input functions (**Figure 4E**).

Next, we plotted the probability of division as a function of density and compared it to our experiments (**Figure 4F**). The probability of division for winner cells was high at low densities and matched that measured in experiments. As density increased, the probability of division decreased. In contrast, loser cells had a lower probability of division that matched experimental measurements. This probability of division is not explicitly implemented in our model and therefore represents an emergent property.

The figures showing the cumulative rates of death and division in our previous publication (Bove et al., MBoC, 2017) were an example from one experiment meant as an illustration. As cell seeding is random in experiments and simulations, this introduces a high degree of variability in the timings of death and division. Therefore, we averaged the cumulative deaths and division counts over 8 experiments and 8 simulations. This revealed a good match between experiments and simulation at most time points (**Figure 4—figure supplement 1B-C**).

4. Most of the time, the authors mention the disappearance of the loser cells in the text, however most of the simulations finish before full disappearance of the loser cells (e.g. Figure 6C, k=0.1 and 1). Is this a matter of time (longer simulation would lead to full disappearance) or is there a steady state with loser cells maintained at low number ?

This is a very interesting point and we thank the reviewers for raising it. We have now run some representative simulations for up to 200h and present these in the main figures and supplementary information (**Figure 6D-E, Figure 1—figure supplement 5B, Figure 5-supplement 1D-E).**

For simulations examining the role of stiffness in mechanical competition, we find that, when the stiffness of loser cells is smaller than that of winners, losers are eliminated within the 200h time frame. When losers were as stiff or stiffer than winners, loser numbers declined at a constant rate, likely as a consequence of their slower growth rate (Figure 5—figure supplement 1E). When we varied the contact inhibition parameter k in winner cells, we found that for k smaller than 0.1, cells were eliminated within 200h. Interestingly, for k=0.1 and above, cells appeared to reach a steady-state, suggesting a true state of coexistence (Figure 5—figure supplement 1D). Finally, in conditions when we vary the initial arrangement of the cells in mechanical competition, we find that cells are completely eliminated by ~150h. Thus, there appear to be some true states of coexistence in mechanical competition.

In models of biochemical competition, with the steepness chosen for the loser cells in the probability of apoptosis function, there appeared to only be coexistence in the fully sorted condition over the durations examined in our study (Figure 6D, Figure 1—figure supplement 5B, Figure 6—figure supplement 1). However, even in the fully sorted condition, the number of loser cells appeared to gradually decrease and the interface between winners and losers slowly shifted over time indicating a slow but constant loss of losers (Video 10). This can be understood from the following mathematical argument as outlined in the next response. Consider a cluster of N loser cells, the rate of apoptosis will scale with the perimeter of the cluster (as ~N^1/2^) while the division rate scales with the area (~N). The change in cell number is of the form dNdt=kdivN−kapoN1/2 where the critical size Ncrit∼(kapokdiv)2 is an unstable equilibrium. Indeed, if N is larger than N_crit_, the loser cells will win the competition, while if N is smaller than N_crit_, the winners will.

5. As stated and shown by the authors, the size of the cluster of loser cells strongly influence the outcome of biochemical competition. It is striking that for the fully sorted conditions, the losers survive irrespective of J-heterotipic. Intuitively this might be related to a perimeter other area ratio which reaches a critical value where apoptosis rate / cell splitting rate (both scaling with perimeter) are always lower than proliferation (scaling with area). Do you also observe such critical cluster size appearing in the partially sorted conditions ? This may be reflected by the final distribution of loser cell-clusters in the partially sorted condition (all clusters being larger than this critical value).

This is a very interesting point and we thank the reviewers for raising it. To answer this question, we have now carried out new simulations in which we systematically vary the initial size of a single loser cell cluster surrounded by winner cells. This revealed that the time to elimination of loser cells increased with loser colony size (Figure 6F).

Because of the increasing demands on computational time, we were not able to ascertain whether elimination time continues to scale exponentially or becomes sigmoidal. However, in videos of fully sorted biochemical competition, we noticed that the position of the interface between winner and loser cells gradually shifts indicating a slow but constant loss of losers (Video 10). This suggested that, with the parameters of our simulation, a true coexistence does not exist. In addition, theoretical consideration suggest that any potential equilibrium will be unstable. Indeed, if we consider a cluster of N loser cells, the rate of apoptosis will scale with the perimeter (as ~N^1/2^) while the division rate scales with the area (~N). The change in cell number is of the form dNdt=kdivN−kapoN1/2 where the critical size Ncrit∼ is an unstable equilibrium. Indeed, if N is larger than N_crit_, the loser cells will win the competition, while if N is smaller than N_crit_, the winners will. However, we note that the observation of coexisting populations in vivo indicates that coexistence states exist and this may result from the action of signalling mechanisms not implemented in our simple simulations.

We agree with the reviewer that it is surprising that J_heterotypic_ does not seem to lead to an acceleration in the kinetics of competition in fully sorted conditions. This may be because, with the range of parameters chosen for J_heterotypic_, the increase in surface of contact between winners and losers only increases the probability of apoptosis of losers very little in fully sorted conditions.